# The ribosomal RNA m$^5$C methyltransferase NSUN-1 modulates healthspan and oogenesis in *Caenorhabditis elegans*

Clemens Heissenberger[1], Jarod A Rollins[2], Teresa L Krammer[1], Fabian Nagelreiter[1], Isabella Stocker[1], Ludivine Wacheul[3], Anton Shpylovyi[1], Koray Tav[1], Santina Snow[2], Johannes Grillari[1,4], Aric N Rogers[2], Denis L J Lafontaine[3], Markus Schosserer[1,2]*

[1]Institute of Molecular Biotechnology, University of Natural Resources and Life Sciences, Vienna, Vienna, Austria; [2]MDI Biological Laboratory, Bar Harbor, United States; [3]RNA Molecular Biology, Fonds de la Recherche Scientifique (F.R.S./FNRS), Université Libre de Bruxelles (ULB), Charleroi, Belgium; [4]Ludwig Boltzmann Institute of Experimental and Clinical Traumatology, Vienna, Austria

**Abstract** Our knowledge about the repertoire of ribosomal RNA modifications and the enzymes responsible for installing them is constantly expanding. Previously, we reported that NSUN-5 is responsible for depositing m$^5$C at position C2381 on the 26S rRNA in *Caenorhabditis elegans*. Here, we show that NSUN-1 is writing the second known 26S rRNA m$^5$C at position C2982. Depletion of *nsun-1* or *nsun-5* improved thermotolerance and slightly increased locomotion at midlife, however, only soma-specific knockdown of *nsun-1* extended lifespan. Moreover, soma-specific knockdown of *nsun-1* reduced body size and impaired fecundity, suggesting non-cell-autonomous effects. While ribosome biogenesis and global protein synthesis were unaffected by *nsun-1* depletion, translation of specific mRNAs was remodeled leading to reduced production of collagens, loss of structural integrity of the cuticle, and impaired barrier function. We conclude that loss of a single enzyme required for rRNA methylation has profound and highly specific effects on organismal development and physiology.

*For correspondence:
markus.schosserer@boku.ac.at

**Competing interests:** The authors declare that no competing interests exist.

## Introduction

Aging is a complex biological process, characterized by progressive aggravation of cellular homeostasis defects and accumulation of biomolecular damages. According to the 'disposable soma theory' of aging, organisms may invest energy either in reproduction or in somatic maintenance (*Kirkwood and Holliday, 1979*). This explains why most lifespan-extending interventions come at the cost of decreased fecundity. De novo protein synthesis by ribosomes, the most energy-demanding process in living cells, affects the balance between aging and reproduction. In fact, reduced overall protein synthesis was shown to extend lifespan in several model organisms, including the nematode *Caenorhabditis elegans* (*Hansen et al., 2007*; *Pan et al., 2007*; *Syntichaki et al., 2007*). Although some evidence indicates that a link between ribosome biogenesis and gonadogenesis in *C. elegans* may exist (*Voutev et al., 2006*), the precise relationship between these pathways in multicellular organisms is still poorly understood. It is conceivable that the optimal function of ribosomes, which requires the presence of ribosomal RNA (rRNA) and ribosomal protein (r-protein) modifications, is monitored by the cell at several stages during development. Thus, introduction of

these modifications might participate in the control of cell fate and cell-cell interactions during development (*Hokii et al., 2010*; *Voutev et al., 2006*).

Eukaryotic ribosomes are composed of about 80 core r-proteins and four different rRNAs, which together are assembled into a highly sophisticated nanomachine carrying the essential functions of mRNA decoding, peptidyl transfer and peptidyl hydrolysis (*Ban et al., 2014*; *Natchiar et al., 2017*; *Penzo et al., 2016*; *Sharma and Lafontaine, 2015*; *Sloan et al., 2017*). Until recently, ribosomes were considered as static homogenous ribonucleoprotein complexes executing the translation of cellular information from mRNA to catalytically active or structural proteins. However, mounting evidence suggests the possibility of ribosomes being heterogeneous in composition with the possibility that some display differential translation with distinct affinity for particular mRNAs (*Genuth and Barna, 2018*). Such heterogeneity in composition may originate from the use of r-protein paralogs, r-protein post-translational modifications, or rRNA post-transcriptional modifications. Indeed, around 2% of all nucleotides of the four rRNAs are decorated with post-transcriptional modifications. These modifications are introduced by specific enzymes such as dyskerin and fibrillarin and guided by specific small nucleolar RNAs (snoRNA) (*Penzo et al., 2016*; *Sloan et al., 2017*). The most abundant rRNA modifications are snoRNA-guided 2′-O-methylations of nucleotide ribose moieties and isomerization of uridine to pseudouridine ($\Psi$). However, some base modifications, which occur less frequently than 2'-O-methylations of ribose and pseudouridines, are installed by specific enzymes, which were largely assumed to be stand-alone rRNA methyltransferases. One exception is the acetyltransferase Kre33 (yeast)/NAT10 (human), which is guided by specialized box C/D snoRNPs (*Sharma et al., 2015*; *Sleiman and Dragon, 2019*). Most of these base modifications are introduced at sites close to the decoding site, the peptidyl transferase center, or the subunit interface. Intriguingly, prokaryotes and eukaryotes share the majority of modifications located in the inner core of the ribosome (*Natchiar et al., 2017*).

In eukaryotes inspected so far, the large ribosomal subunit contains two m$^5$C residues. This is notably the case in budding yeast (on 25S rRNA), in the nematode worm (26S) and in human cells (28S) (*Sharma and Lafontaine, 2015*). Rcm1/NSUN-5, an enzyme of the NOP2/Sun RNA methyltransferase family, is responsible for introducing m$^5$C at residue C2278 and C2381 on 25S/26S rRNA in yeast and worms, respectively (*Gigova et al., 2014*; *Schosserer et al., 2015*; *Sharma et al., 2013*). Recently, our group and others identified the conserved target cytosines in humans and mice, C3782, and C3438, respectively (*Janin et al., 2019*; *Heissenberger et al., 2019*). We also reported that lack of this methylation is sufficient to alter ribosomal structure and ribosome fidelity during translation, while extending the lifespan and stress resistance of worms, flies and yeast (*Schosserer et al., 2015*). However, the identity of the second worm m$^5$C rRNA methyltransferase remains unknown.

Here, we report that NSUN-1 is responsible for writing the second *C. elegans* 26S m$^5$C (position C2982). We then investigate the physiological roles of NSUN-1, comparing them systematically to those of NSUN-5. We show that NSUN-1 and NSUN-5 distinctly modulate fundamental biological processes such as aging and fecundity. In particular, depletion of *nsun-1* impairs fecundity, gonad maturation and remodels translation of specific mRNAs leading to cuticle defects. We conclude that loss of NSUN-1 introducing a single rRNA modification is sufficient to profoundly and specifically alter ribosomal function and, consequently, essential cellular processes.

## Results

### NSUN-1 is responsible for writing m$^5$C at position C2982 on *C. elegans* 26S rRNA

Previously, we showed that an m$^5$C modification is introduced at position C2381 on the 26S rRNA of *C. elegans* large ribosomal subunit by NSUN-5 (*Adamla et al., 2019*; *Schosserer et al., 2015*), which is required to modulate animal lifespan and stress resistance (*Schosserer et al., 2015*). On this basis, we were interested to learn if other related rRNA methyltransferases in *C. elegans* might display similar properties.

Therefore, we investigated the RNA substrate of NSUN-1 (also formerly known as NOL-1, NOL-2, or W07E6.1) and its potential roles in worm development and physiology. NSUN-1 is a member of the NOP2/Sun RNA-methyltransferase family. Since there are only two known m$^5$C residues on

worm 26S rRNA (*Sharma and Lafontaine, 2015*; *Trixl and Lusser, 2019*), one of them at C2381, being installed by NSUN-5, we speculated that NSUN-1 might be required for introducing the second m5C residue at position C2982. Notably, both 26S m5C sites are localized close to the decoding site and peptidyl transferase center of the ribosome, and are highly conserved between yeast, worm and human (*Figure 1A,B*).

In order to test if NSUN-1 is involved in large ribosomal subunit m5C methylation, we first sought to identify a suitable model to study loss of NSUN-1. We selected the *tm6081* allele which has a deletion in the 3' untranslated region (3' UTR) of *nsun-1* (*Figure 1—figure supplement 1A*). After letting single hermaphrodites, which were heterozygous for *tm6081* self-fertilize, we were unable to detect any viable offspring carrying the homozygous mutation (*Figure 1—figure supplement 1B*). This indicates that *tm6081* is a recessive lethal mutation, which agrees with previous reports about lethality of *nsun-1* depletion by egg-onset RNAi (*Kamath et al., 2003*; *Piano et al., 2002*). Since *nsun-1* mRNA levels were only decreased by 40% in animals heterozygous for *tm6081* (*Figure 1—figure supplement 1C*), we decided to use RNAi instead with a chance to achieve higher knockdown efficiencies. As it will become evident below, there are several other advantages of using RNAi in *C. elegans*. One is that it allows to deplete a factor of interest at a particular life stage only (e.g. in adult worms), another is that it allows performing tissue-specific knockdown of gene expression.

26S rRNA was purified from worms treated with siRNAs specific to NSUN-1-encoding mRNAs on sucrose gradients, digested to single nucleosides, and analyzed by quantitative HPLC. In our HPLC assay, the m5C nucleoside eluted at 12 min, as established with a m5C calibration control (data not shown). The depletion of NSUN-1 was conducted in two genetic backgrounds: N2 (wildtype), and NL2099 (an RNAi-hypersensitive strain due to mutation in *rrf-3*) (*Figure 1—figure supplement 2*). Treating N2 worms with an empty control vector, not expressing any RNAi, did not significantly reduce the levels of 26S rRNA m5C methylation (*Figure 1C*, 97% instead of 100%). Interestingly, treating N2 worms with an RNAi construct targeting *nsun-1* led to a reduction of 26S rRNA m5C methylation by 35% (*Figure 1C*). In the NL2099 strain, *nsun-1* RNAi treatment also led to a reduction of 26S rRNA m5C methylation by 26% (*Figure 1D*). A second independent biological replicate confirmed these findings (*Figure 1—figure supplement 3*).

As there are only two known modified m5C residues on worm 26S rRNA, and since one of them is introduced by NSUN-5 (*Adamla et al., 2019*; *Schosserer et al., 2015*), a complete loss of NSUN-1 activity was expected to result in a 50% decrease in m5C methylation. However, protein depletion achieved with RNAi is usually not complete. It is not clear why the level of m5C depletion was not higher in the RNAi-hypersensitive strain in comparison to the N2 strain; nonetheless, RNAi-mediated depletion of *nsun-1* significantly reduced the levels of 26S rRNA m5C modification in both worm strains, thus, we conclude that NSUN-1 is responsible for 26S rRNA m5C methylation.

We analyzed the 26S rRNA m5C levels in a *nsun-5* deletion strain as control (strain JGG1, *Figure 1E*). In this case, we observed a near 2-fold reduction in methylation (58% residual), as expected from the known involvement of NSUN-5 in modification at position C2381. When *nsun-1* was additionally depleted by RNAi in the *nsun-5* knockout animals, the level of 26S rRNA m5C was further reduced to 43%, again in agreement with our conclusion that NSUN-1 is responsible for methylating the second position, C2982.

Since our conclusion is based on depletion of *nsun-1* to ~20% residual expression and not on a full gene knockout (*Figure 1—figure supplement 2*), we cannot exclude the formal possibility that NSUN-1 might not be the only m5C2982 writer in *C. elegans*. However, we consider this possibility to be highly unlikely because the combination of a knockout of Rcm1 with a catalytic mutation of Nop2 in yeast was sufficient to completely remove m5C from 25S rRNA (*Sharma et al., 2013*).

To further prove that NSUN-1 is not involved in C2381 modification, methylation levels at this position were specifically tested by Combined Bisulfite Restriction Analysis (COBRA) assay in animals depleted of *nsun-1* or *nsun-5*. This method is based on bisulfite conversion of total RNA, followed by PCR amplification and restriction digest, yielding two bands in case of methylation at C2381 and three bands in case of non-methylation (*Adamla et al., 2019*). As expected, only *nsun-5* depletion strongly reduced methylation at C2381, and there was no residual m5C2381 in the *nsun-5* knockout strain, while *nsun-1* RNAi had no effect on modification at this position (*Figure 1F*). Bisulfite sequencing is well-known to be sensitive to RNA secondary structure (*Warnecke et al., 2002*), which

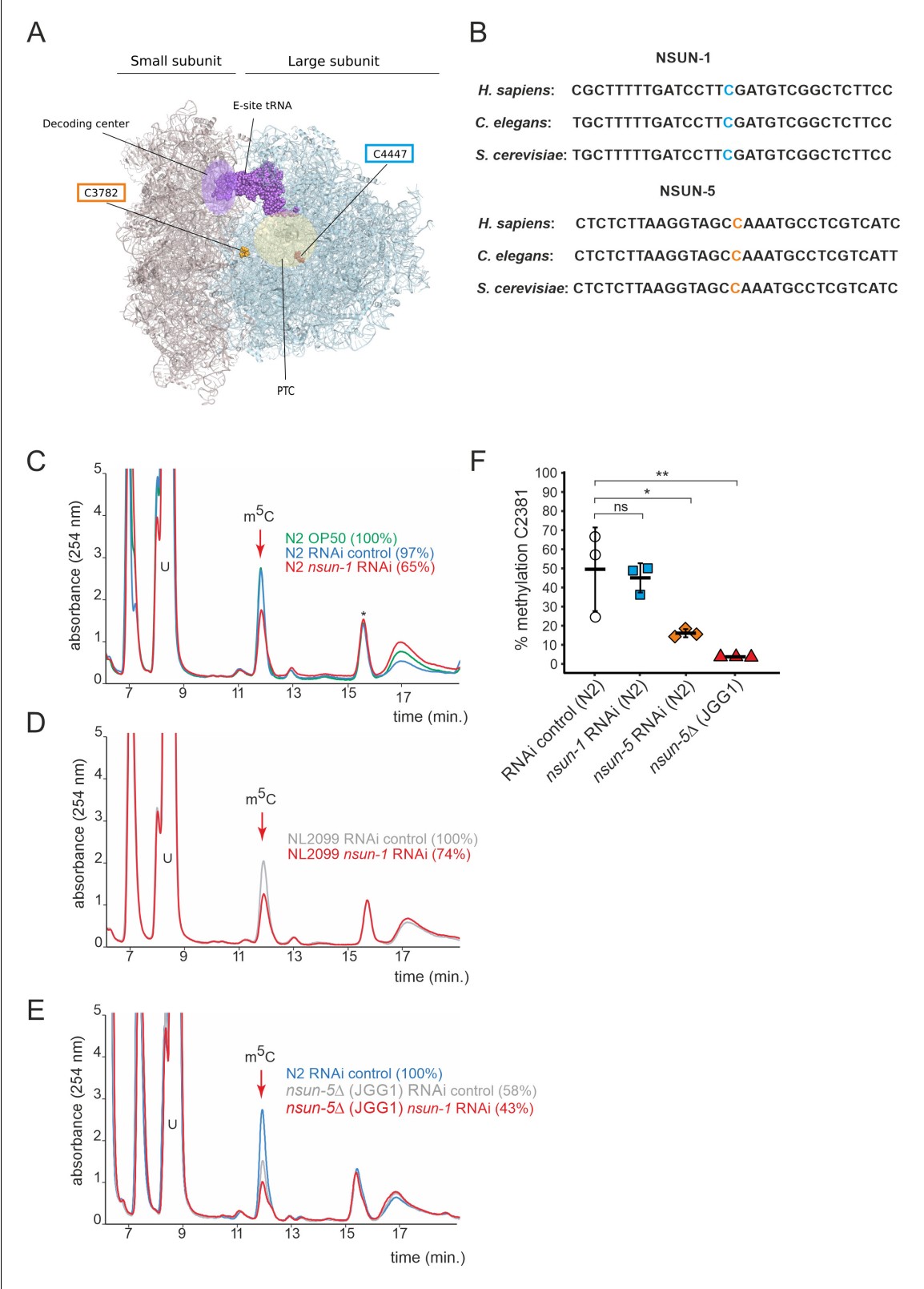

**Figure 1.** NSUN-1 is responsible for large ribosomal subunit 26S rRNA m$^5$C methylation. (**A**) Location of the two eukaryotic large ribosomal subunit m$^5$C residues within the 3D structure of the human ribosome. For reference, important functional sites are indicated (DCS = decoding site, PTC = peptidyl transferase center). In *C. elegans*, NSUN-1 is responsible for m$^5$C2982 (this work) while NSUN-5 installs m$^5$C2381 (***Schosserer et al., 2015***). (**B**) Regions surrounding the sites modified by NSUN-1 and NSUN-5 are evolutionarily conserved between yeast, worms, and humans. The

*Figure 1 continued on next page*

*Figure 1 continued*

modified cytosine is indicated. (C–E) Purified 26S rRNA was isolated by sucrose gradient centrifugation, digested to single nucleotides and analyzed by quantitative HPLC. *nsun-1* knockdown consistently leads to a decrease of m⁵C levels. (C) N2 worms were analyzed as either: untreated (OP-50), treated with an RNAi control or with a *nsun-1* targeting RNAi. (D) NL2099 RNAi-hypersensitive worms were treated with the RNAi control or with the *nsun-1* targeting RNAi. (E) N2 strain treated with RNAi control and the *nsun-5* deletion strain (JGG1) treated with control RNAi or a *nsun-1* targeting RNAi. For quantification of m⁵C peak area, the peak was normalized to the peak eluting at 16 min (asterisk). The experiment was independently repeated once with similar outcome (see *Figure 1—figure supplement 3*). (F) Quantification of the enzymatic activity of NSUN-5 using the COBRA assay for N2 worms, subjected to either *nsun-5* or *nsun-1* RNAi, and the *nsun-5* mutant strain JGG1 (*nsun-5Δ*). Loss of *nsun-5* leads to significantly decreased methylation levels at C2381, whereas *nsun-1* RNAi does not alter methylation at this site (three independent biological replicates, one-way ANOVA with Dunnett´s post test, α = 0.05, *p<0.05, **p<0.01).

The online version of this article includes the following source data and figure supplement(s) for figure 1:

**Source data 1.** Raw data of COBRA assays to quantify methylation levels.
**Figure supplement 1.** Characterization of the *tm6081* allele.
**Figure supplement 2.** RNAi effectively depletes *nsun-1* in *C.elegans*.
**Figure supplement 3.** NSUN-1 methylates 26S rRNA.

likely explains why, despite repeated attempts, we could not monitor modification at position C2982 by use of this technique.

In conclusion, NSUN-1 and NSUN-5 are each responsible for installing one m⁵C onto the worm 26S rRNA, with NSUN-1 being responsible for position C2982 and NSUN-5 for position C2381 under the *bona fide* assumption that indeed only two m⁵C positions are present as described (*Sharma and Lafontaine, 2015*).

## The soma-specific depletion *of nsun-1* extends lifespan

Next, we investigated if knockdown of *nsun-1* modulates healthy lifespan in a similar fashion as that described for *nsun-5* (*Schosserer et al., 2015*). In order to achieve this aim, we depleted *nsun-1* by RNAi in N2 wild-type animals starting from day 0 of adulthood and, quite surprisingly, we did not observe any extension of mean or maximum lifespan (*Figure 2A*, *Table 1*). Next, we evaluated the stress resistance of adult worms upon *nsun-1* or *nsun-5* depletion, as an increased health at an advanced age often improves resilience to adverse events (*Lithgow et al., 1994*). Indeed, depletion of either *nsun-1* or *nsun-5* increased resistance to heat stress compared to the RNAi control (*Figure 2B*). Furthermore, we tracked the movement of animals treated with either an empty vector control or two constructs expressing RNAi directed against *nsun-1* or *nsun-5* in a time course analysis, starting at day 1 of adulthood up to day 16. Interestingly, we observed increased average speed at day 8 of adulthood in both *nsun-1* (+47.8%, p=0.009) and *nsun-5* (+34.7%, p=0.073) depleted animals compared to the control, as well as at day 12 (*nsun-1* RNAi: +10.2%, p=0.539; *nsun-5* RNAi: +73.5%, p=0.008 compared to the control) (*Figure 2C*). Other timepoints remained unaffected. Thus, while *nsun-1* knockdown does not extend lifespan, it improves two healthspan parameters, namely thermotolerance and midlife locomotion (*Bansal et al., 2015*; *Rollins et al., 2017*).

Intrigued that depletion of *nsun-1* did not extend the lifespan of *C. elegans* in a similar fashion as that reported for *nsun-5* when whole adult animals were treated with RNAi, we reasoned that performing tissue-specific depletion of *nsun-1* might help us to further elucidate a possible effect on lifespan. We focused on the comparison of the germline and somatic tissues, because somatic maintenance and aging are evolutionarily tightly connected (*Kirkwood and Holliday, 1979*), and because signals from the germline modulate *C. elegans* lifespan (*Hsin and Kenyon, 1999*). In addition, only loss of soma- but not germline-specific eIF4E isoforms, which are central regulators of cap-dependent translation, extend nematode lifespan (*Syntichaki et al., 2007*). To test if *nsun-1* has similar tissue-specificity, we made use of worm strains sensitive to RNAi only in either the germline or somatic tissues. This is achieved, on the one hand by mutation of *rrf-1*, which is required for amplification of the dsRNA signal in the somatic tissues (*Kumsta and Hansen, 2012*; *Sijen et al., 2001*), and, on the other hand by functional loss of the argonaute protein *ppw-1* rendering the germline resistant to RNAi (*Tijsterman et al., 2002*). Interestingly, germline-specific *nsun-1* RNAi had no effect on animal lifespan (*Figure 2D*, *Table 1*), but depletion of *nsun-1* in somatic tissues reproducibly increased mean lifespan by ~10% (*Figure 2E*, *Table 1*).

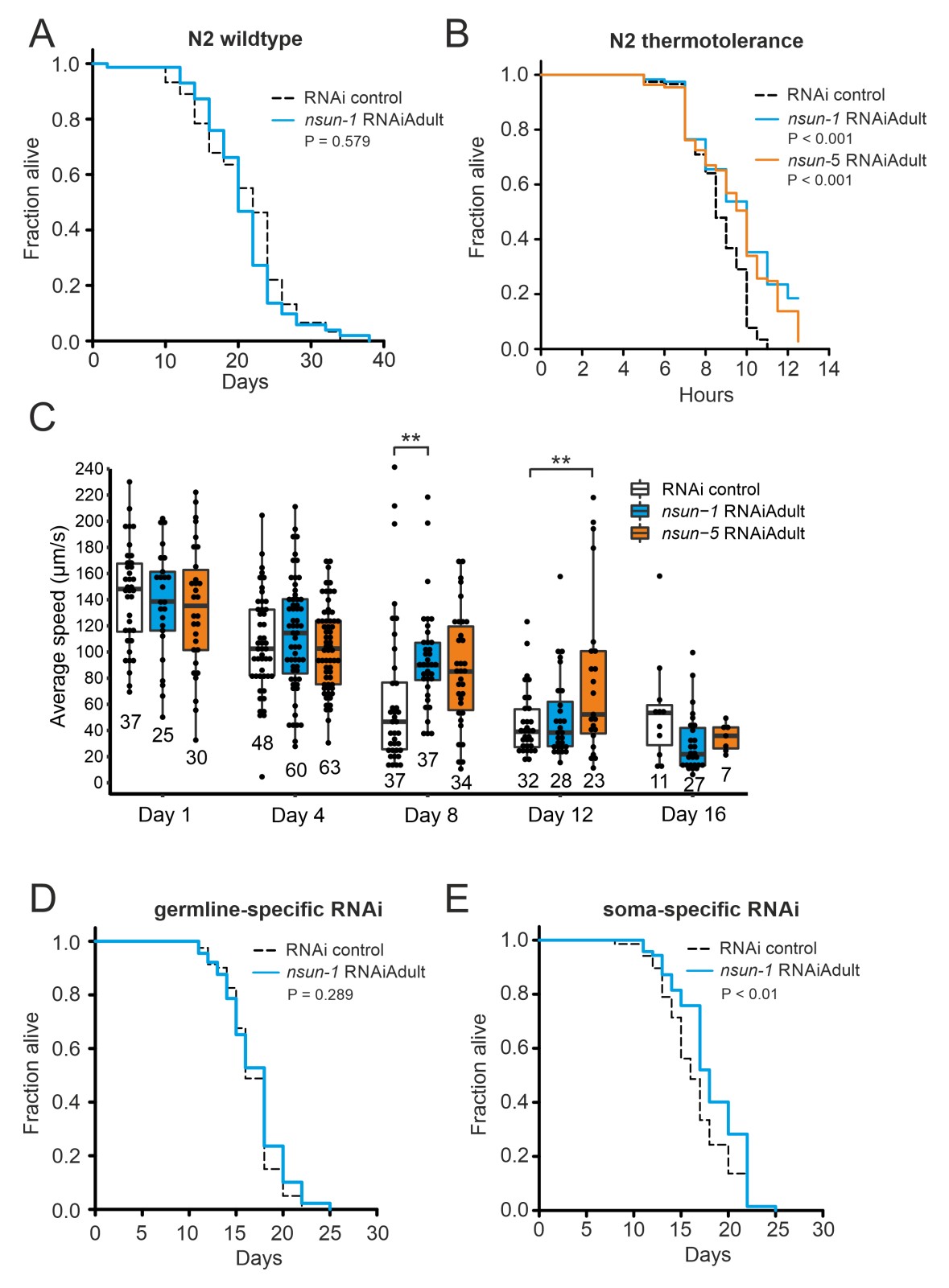

**Figure 2.** The soma-specific depletion of *nsun-1* extends lifespan. (**A**) *Nsun-1* whole-body adult-onset RNAi (N2 wildtype strain) does not affect lifespan. Three independent biological experiments were performed. One representative replicate it shown. n = 75 animals per condition, log-rank test, not significant. (**B**) N2 wildtype animals treated with either *nsun-1* or *nsun-5* RNAi and subjected to heat stress (35°C) show increased survival compared to the RNAi control. Nine pooled biological replicates are shown. Pooled n ≥ 100 animals per condition, log-rank, p<0.001. (**C**) Average speed [μm/s] of

*Figure 2 continued on next page*

*Figure 2 continued*

N2 wildtype worms as indicator of the health status was measured at day 1, 4, 8, 12, and 16 of adulthood. Movies of animals treated with either RNAi control, *nsun-1* or *nsun-5* RNAi were recorded. One representative experiment is shown. Three biological replicates were performed with similar outcome. n ≥ 20 animals per condition at day 1. The black line indicates median. Statistical significance at each timepoint was determined using multiple comparison adjusted t-tests by the Holm-Sidak method. α = 0.05, **p<0.01. (D–E) Lifespan analysis of germline- (NL2098) and soma-specific RNAi strains (NL2550). Worms were treated with either RNAi control or *nsun-1* RNAi (adult-onset). Only soma-specific knockdown of *nsun-1* results in increased lifespan (E) while germline-specific knockdown does not (D). Two independent biological experiments were performed. One representative replicate it shown. n(NL2098)=90 animals per condition and replicate, log-rank, not significant, n(NL2550)=90 animals per condition and replicate, log-rank, (p<0.01). A summary table of the individual replicates of lifespan experiments is provided as *Table 1*.

The online version of this article includes the following source data for figure 2:

**Source data 1.** Raw data of lifespan, motility, and thermotolerance experiments.

In conclusion, both NSUN-1 and NSUN-5 m⁵C rRNA methyltransferases mildly affect thermotolerance and mobility of wild-type nematodes at midlife. Whole-animal *nsun-5* depletion expands mean lifespan by 17% (*Schosserer et al., 2015*), and in contrast to this, a 10% lifespan extension is only detected after depletion of *nsun-1* specifically in the somatic tissues.

## The somatic tissue-specific depletion of *nsun-1* affects body size, fecundity, and gonad maturation

The 'disposable soma theory' of aging posits that long-lived species exhibit impaired fecundity and reduced number of progenies. The proposed underlying cause is that energy is invested in the maintenance of somatic tissues rather than in rapid reproduction (*Kirkwood and Holliday, 1979*). In keeping with this theory, we expected that the absence of *nsun-1* in somatic tissues, which increased longevity, may reduce fecundity. Therefore, we measured the brood size upon *nsun-1* and *nsun-5* depletion by RNAi. After reaching adulthood but prior to the egg-laying stage, worms were transferred to individual wells of cell culture plates containing NGM-agar and fed with bacteria expressing the specific RNAi or, as control, the empty vector. Egg production was impaired upon *nsun-1* knockdown (reduced by 42%), but this was not the case upon *nsun-5* knockdown (reduced by 2%) (*Figure 3A*). Egg production ceased rapidly after day one in all conditions (*Figure 3—figure supplement 1A*).

**Table 1.** Summary of individual lifespan and thermotolerance experiments.

| strain | treatment | replicate | mean survival | s.d. | dead/total | P-value |
|--------|-----------|-----------|---------------|------|------------|---------|
| N2 | RNAi control | 1 | 20.8 days | ±0.9 | 47/75 | 0.579 |
| N2 | *nsun-1* RNAi | 1 | 20.7 days | ±0.8 | 52/75 | |
| N2 | RNAi control | 2 | 18.9 days | ±0.6 | 75/90 | 0.694 |
| N2 | *nsun-1* RNAi | 2 | 18.9 days | ±1.1 | 74/90 | |
| N2 | RNAi control | 3 | 20.7 days | ±0.3 | 87/90 | 0.474 |
| N2 | *nsun-1* RNAi | 3 | 21.0 days | ±0.4 | 82/90 | |
| NL2550 | RNAi control | 1 | 16.5 days | ±0.4 | 66/90 | 0.009 |
| NL2550 | *nsun-1* RNAi | 1 | 18.0 days | ±0.4 | 68/90 | |
| NL2550 | RNAi control | 2 | 18.6 days | ±0.3 | 83/90 | <0.001 |
| NL2550 | *nsun-1* RNAi | 2 | 20.2 days | ±0.4 | 75/90 | |
| NL2098 | RNAi control | 1 | 16.7 days | ±0.3 | 80/90 | 0.289 |
| NL2098 | *nsun-1* RNAi | 1 | 17.0 days | ±0.3 | 89/90 | |
| NL2098 | RNAi control | 2 | 19.2 days | ±0.3 | 87/90 | 0.068 |
| NL2098 | *nsun-1* RNAi | 2 | 18.4 days | ±0.3 | 88/90 | |
| N2 | heat/RNAi control | pool | 8.6 hr | ±0.1 | 117/117 | control |
| N2 | heat/*nsun-1* RNAi | pool | 12.5 hr | ±0.7 | 97/119 | <0.001 |
| N2 | heat/*nsun-5* RNAi | pool | 9.7 hr | ±0.3 | 106/109 | <0.001 |

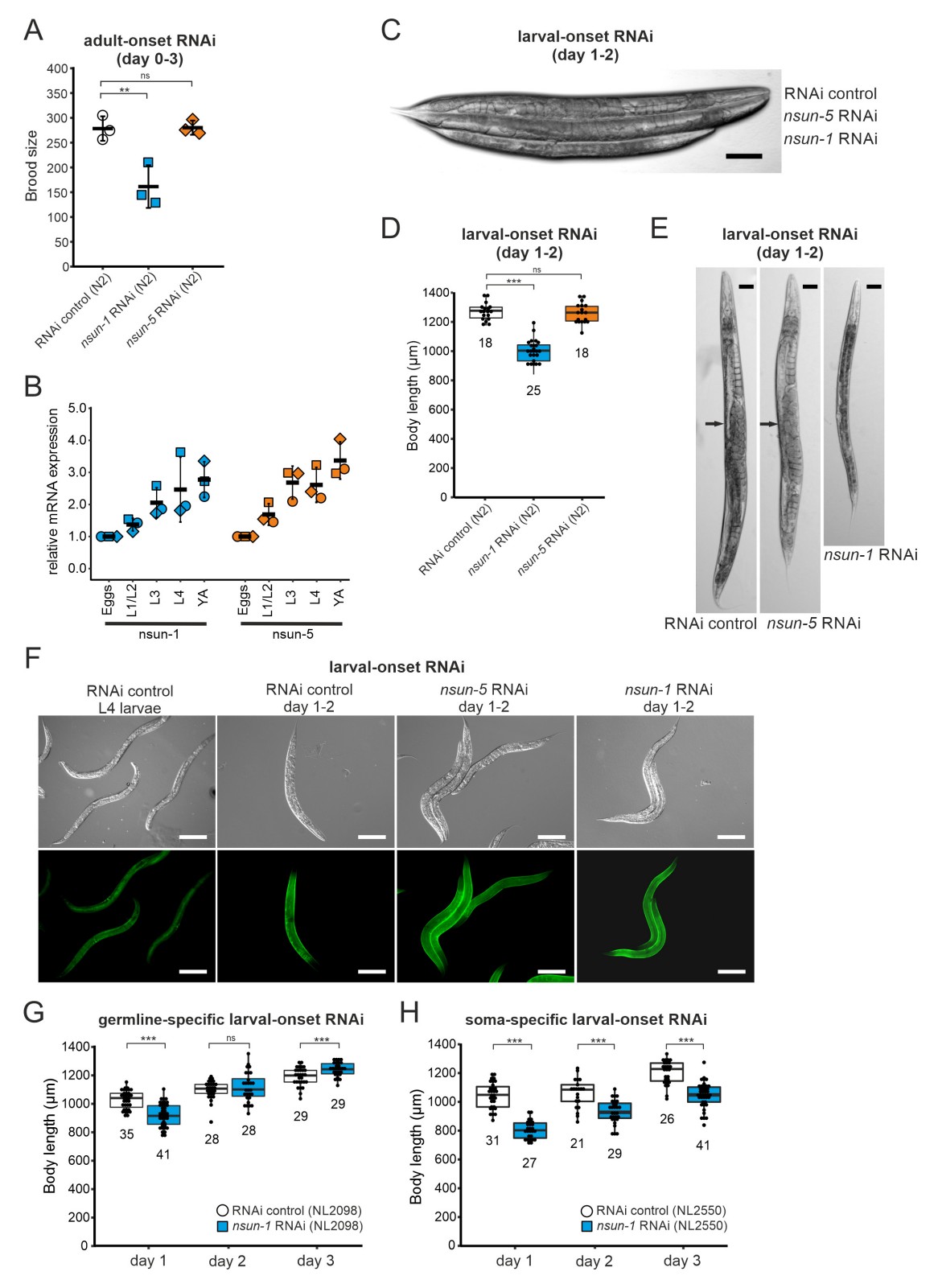

**Figure 3.** Loss of *nsun-1* reduces body size and impairs fecundity. (**A**) Brood size analysis of adult-onset RNAi exposed animals at day 0–3 of adulthood (=day 3–6 from egg). Eggs of individual worms were counted and the total number of eggs per worm is shown. Knockdown of *nsun-1* but not *nsun-5* induced a significant reduction in brood size compared to control RNAi (three independent experiments, n = 5 per condition and per experiment, one-way ANOVA with Dunnett´s post test, α = 0.05, **p<0.01). Error bars indicate standard deviation. (**B**) RT-qPCR analysis of wild-type animals at different

*Figure 3 continued on next page*

*Figure 3 continued*

stages of development (eggs, L1/L2 larvae, L3 larvae, L4 larvae, and young adults). *tba-1* was used for normalization and expression is shown relative to eggs. Error bars represent standard deviation of three biological replicates, one-sample t-test against expected value of 1 with multiple comparison correction by Holm's method did not reveal significant differences. (C) Representative DIC images of larval-onset RNAi exposed nematodes at day 1–2 of adulthood show that only *nsun-1* but not *nsun-5* RNAi decreased the body length and altered general morphology compared to the RNAi control. Scale bar, 100 μm. (D) Quantification of mean body length of 1–2 day old larval-onset RNAi exposed adult worms. The body size of *nsun-1* RNAi-treated worms was significantly reduced compared to the RNAi control and *nsun-5* RNAi. The experiment was independently performed two times and one representative replicate is shown. n(RNAi control)=18, n(*nsun-1* RNAi)=25, n(*nsun-5* RNAi)=19, one-way ANOVA with Dunnett's post, α = 0.05, ***p<0.001. Error bars represent standard deviation. (E) Larval-onset *nsun-1* RNAi-treated adults at day 1–2 of adulthood had reduced body size and lacked embryos (arrow). Scale bar, 50 μm. (F) Loss of *nsun-1* did not impair expression of the adult-specific marker COL-19::GFP. The TP12 strain was used and young adult animals (day 1–2 of adulthood) treated with either larval-onset control RNAi, *nsun-1* RNAi or *nsun-5* RNAi were imaged in DIC and fluorescent mode. L4 control RNAi worms, which did not express GFP specifically in the hypodermis, were used as negative control. Scale bar, 200 μm. (G–H) Soma- but not germline-specific *nsun-1* larval-onset RNAi phenocopied the mean body length defect upon whole-body *nsun-1* knockdown. The germline-specific NL2098 strain (G) and the soma-specific NL2550 (H) strain were used and measured on three consecutive days after reaching adulthood. n ≥ 21 for each day and condition. Two independent experiments were performed and one representative replicate is shown. Two-tailed t-test, ***p<0.001. Error bars represent standard deviation.

The online version of this article includes the following source data and figure supplement(s) for figure 3:

**Source data 1.** Raw data of brood size, expression during development and body length experiments.
**Figure supplement 1.** *nsun-1* depleted worms display impaired fecundity.

Thus far, all the experiments were performed on worms subjected to adult-onset *nsun-1* knockdown, as animals depleted of *nsun-1* during development were smaller and were infertile upon adulthood. To follow up on these observations, we measured mRNA expression levels of both m$^5$C rRNA methyltransferases at different developmental stages including eggs, L1/L2 larvae, L3 larvae, L4 larvae, and young adults. RT-qPCR indicated that both *nsun-1* and *nsun-5* mRNA levels constantly increase during development (*Figure 3B*). The same observation applied to mRNA levels of *nsun-2* and *nsun-4* (*Figure 3—figure supplement 1B*), indicating that all four members of the NSUN-protein family might play important roles during development.

To further assess whether *nsun-1* expression is indeed necessary for progressing faithfully through larval stages, we captured images of young adult animals subjected to larval-onset RNAi. The disparity in body size between RNAi control and *nsun-1* RNAi was apparent, whereby *nsun-1* depleted animals showed reduced length by approximately 20%. Interestingly, this reduced body size was not seen upon *nsun-5* RNAi treatment (*Figure 3C,D*).

In addition, we imaged 3 day old animals using differential interference contrast (DIC) microscopy. Worms subjected to *nsun-1* knockdown displayed morphological alterations; specifically, the gonad appeared severely distorted (*Figure 3E*). In contrast, RNAi control and *nsun-5* RNAi showed comparable morphology of distal and proximal gonads (*Figure 3E*). Consequently, we hypothesized that *nsun-1* RNAi-treated worms might be arrested in early L4 larval stage when the gonad is not yet fully developed and animals still grow, instead of normally reaching adulthood after 3 days like RNAi control or *nsun-5* RNAi-treated nematodes. To test this possibility, we used the TP12 *kaIs12[col-19::GFP]* translational reporter strain, which expresses COL-19::GFP specifically upon reaching adulthood, but not during larval stages. Surprisingly, larval-onset RNAi against *nsun-1* or *nsun-5* did not reveal differences in the expression of COL-19::GFP as compared to RNAi control, suggesting that neither *nsun-1* nor *nsun-5* induce larval arrest (*Figure 3F*). Together with the reduced brood size upon adult-onset RNAi, these findings imply that loss of *nsun-1* induces phenotypic changes in the reproductive organs of *C. elegans*.

Since knockdown of *nsun-1* extended lifespan only when it was applied to somatic tissues, we hypothesized that body length might also be affected when these tissues are specifically targeted for depletion. To test this hypothesis, we depleted *nsun-1* specifically in the germline or in somatic tissues using tissue-specific RNAi strains and measured body size during three consecutive days after adulthood was reached. While germline-specific knockdown of *nsun-1* did not induce any changes in body size (*Figure 3G*), soma-specific knockdown revealed a decrease in body length by 23% on day 1, 11% on day 2 and 12% on day 3 compared to the RNAi control (*Figure 3H*), phenocopying *nsun-1* depletion in wild-type animals after whole-animal RNAi. Similarly, no effect of *nsun-1* knockdown

was evident in another germline-specific RNAi strain, which was recently developed to enhance germline-specificity (*Zou et al., 2019*; *Figure 3—figure supplement 1C*).

In conclusion, *nsun-1* but not *nsun-5* depletion impairs body size and morphology of the gonad and leads to a significant reduction of brood size. Furthermore, these phenotypes are also observed when *nsun-1* is specifically knocked-down in somatic tissues, but not when depleted in the germline only.

## NSUN-1 is required for the transition of meiotic germ cells to mature oocytes

To further investigate the mechanisms underlying impaired fecundity upon *nsun-1* knockdown, we analyzed the morphology of the gonad in *nsun-1* depleted animals in more detail. The germline of adult hermaphrodites resides within the two U-shaped arms of the gonad, which contains germ cells at various stages of differentiation (*Figure 4A*). The gonad is sequentially developing from the proliferative germ cells near the distal tip cell, through the meiotic zone into the loop region, finally culminating in fully-formed oocytes in the proximal gonad (*Pazdernik and Schedl, 2013*). The limiting factor for fecundity in self-fertilizing hermaphrodites is sperm produced in the spermatheca (*Hodgkin and Barnes, 1991*).

Upon visualizing the germline cell nuclei with DAPI-staining, no oocytes were observed in worms after knockdown of *nsun-1* in contrast to RNAi control treated animals (*Figure 4B*). The mitotic zone at the distal end of the gonad appeared normal in *nsun-1* depleted animals, whereas oocyte production starting at the pachytene zone was hampered. Analysis of GFP::RHO-1 and NMY-2::GFP expressing worm strains, which specifically express GFP in the germline, confirmed our observations (*Figure 4C*, *Figure 4—figure supplement 1*). The gonads of control and *nsun-5* RNAi-treated animals appeared normal, clearly depicting the different stages of *in-utero* embryo development, whereas the germline of *nsun-1* RNAi-treated animals displayed a strikingly altered morphology. Importantly, this phenotype showed 100% penetrance in worms exposed to *nsun-1* RNAi.

Since other phenotypes observed upon *nsun-1* depletion were detected in soma- but absent from germline-specific RNAi-treated strains, we hypothesized that the somatic part of the gonad might specifically require NSUN-1 for normal oocyte production. Indeed, soma-specific depletion of *nsun-1* phenocopied the distorted gonad morphology of wild-type animals exposed to *nsun-1* RNAi (*Figure 4D*). Remarkably, upon germline-specific knockdown of *nsun-1,* the gonad appeared completely unaffected (*Figure 4E*; *Figure 4—figure supplement 2*).

## NSUN-1 is not essential for pre-rRNA processing and global protein synthesis

Since the only known function of NSUN-1 and NSUN-5 is m$^5$C methylation of rRNA, we reasoned that methylation-induced alterations of ribosome biogenesis and function might explain the observed phenotypes. Therefore, we tested if the presence of NSUN-1 or NSUN-5 is required for ribosomal subunit production and pre-rRNA processing. To this end, total RNA was extracted from worms treated with *nsun-1* RNAi, separated by denaturing agarose gel electrophoresis and processed for northern blot analysis (*Figure 5A,B*). Again, two reference worm strains were used (N2 and NL2099). Upon *nsun-1* knockdown, we observed a mild accumulation of the primary pre-rRNA transcript and of its immediate derivative, collectively referred to as species 'a' (*Figure 5A*; *Bar et al., 2016*; *Saijou et al., 2004*), as well as a mild accumulation of the pre-rRNAs 'b' and 'c'' (*Figure 5A,B*, see lane 1 and 3 as well as 5 and 6). Again, these findings were observed in both worm backgrounds tested, N2 and NL2099.

For comparison, we also analyzed rRNA processing in *nsun-5* deletion worms (JGG1 strain) in presence and absence of NSUN-1 (*nsun-1* RNAi in JGG1). In both cases, we noted an important reduction in the overall production of ribosomal RNAs (*Figure 5B,C*), with an apparent increase of rRNA degradation (seen as an increase in accumulation of metastable RNA fragments, in particular underneath the 18S rRNA). Furthermore, we observed that NSUN-1 is not required for mature rRNA production as shown by the unaffected levels of mature 18S and 26S rRNAs (*Figure 5C*). This was confirmed by determining the 26S/18S ratio, which was 1.0 as expected since both rRNAs are produced from a single polycistronic transcript (*Figure 5C*). The levels of the other two mature rRNAs (5S and 5.8S) were also unaffected (*Figure 5—figure supplement 1*). This behavior was shown in

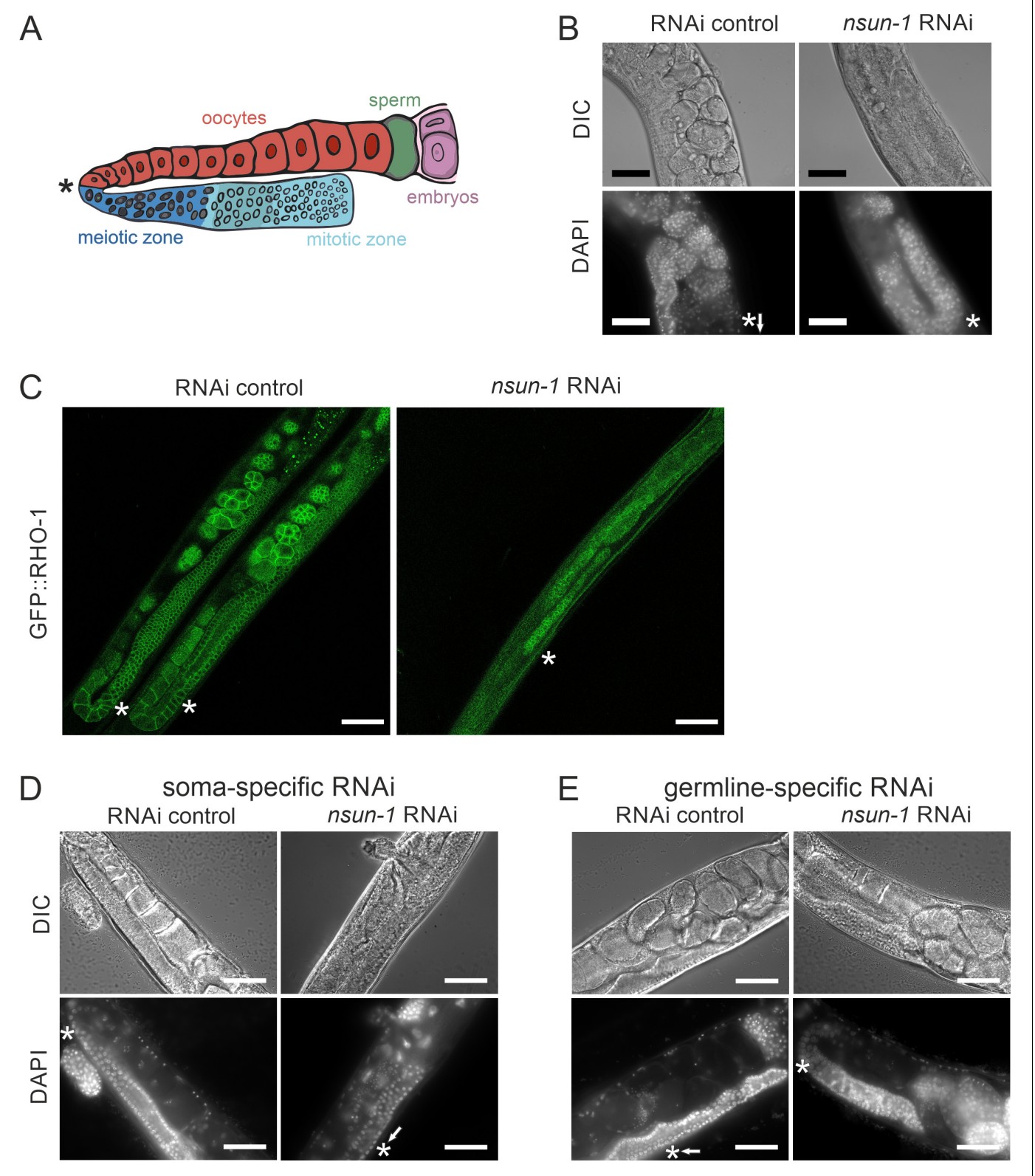

**Figure 4.** Soma-specific depletion of *nsun-1* blocks oogenesis. (**A**) Schematic of one gonad arm in *C. elegans*. Germ cell replication starts in the distal mitotic zone. After passing through the meiotic zone, oocytes further mature and are fertilized by sperm produced in the spermatheca. In panels **A–E**, an asterisk indicates the gonadal region impaired in *nsun-1* RNAi exposed animals. This area corresponds to the transition between the meiotic zone and oocyte maturation. (**B**) Microscopic image of one gonad arm of young adult worms subjected to either control or *nsun-1* RNAi. Worms were

*Figure 4 continued on next page*

*Figure 4 continued*

imaged in DIC mode and nuclei of fixed animals were stained with DAPI. Scale bar, 40 μm. (C) Confocal imaging of the gonad-specific GFP::RHO-1 expressing SA115 strain revealed altered gonad morphology upon *nsun-1* knockdown (see also *Figure 4—figure supplement 1*). Scale bar, 40 μm. Altered gonad morphology was observed in all analyzed animals exposed to *nsun-1* RNAi (n > 50). (D–E) Soma- but not germline-specific *nsun-1* RNAi phenocopied altered gonad morphology upon whole-body *nsun-1* depletion. NL2550 was used for soma- (D) and NL2098 for germline-specific knockdown (E). One gonad arm of one representative 2 day old adult animal was imaged in DIC mode and nuclei were stained with DAPI following fixation. Scale bar, 40 μm.

The online version of this article includes the following figure supplement(s) for figure 4:

**Figure supplement 1.** *nsun-1* but not *nsun-5* depletion inflicts a defect in oogenesis.

**Figure supplement 2.** Somatic- but not germline-specific *nsun-1* depletion causes defective oogenesis.

both worm backgrounds, N2 and NL2099, used. The overall decrease in mature ribosome production observed in *nsun-5* deletion worms did not affect the ratio of mature ribosomal subunits (26S/18S ratio of 1.0) (*Figure 5C*). In agreement with the reduced amounts of 18S and 26S rRNA observed in *nsun-5* deletion worms, total amounts of all precursors detected were reduced (*Figure 5B*). Analysis of low molecular weight RNAs by acrylamide gel electrophoresis revealed the absence of NSUN-5 to severely inhibit processing in the internal transcribed spacer 2 (ITS2), which separates the 5.8S and 26S rRNAs on large precursors. This was illustrated by the accumulation of 3'-extended forms of 5.8S, and of short RNA degradation products (*Figure 5—figure supplement 1*, see lanes 3 and 4). Depletion of *nsun-1* partially suppressed the effect of *nsun-5* deletion: the overall production of mature rRNA and, in particular, the amount of mature 26S rRNA was increased (ratio of 1.2) (*Figure 5C*). Consistently, the accumulation of 3'-extended forms of 5.8S and of short RNA degradation products was reduced (*Figure 5—figure supplement 1*).

In order to test if mature ribosomes of animals lacking any of the two m$^5$C rRNA methyltransferases might be functionally defective, we analyzed global protein synthesis by incorporation of puromycin in N2 worms treated with either RNAi control, *nsun-1* or *nsun-5* RNAi. Worms were exposed to puromycin for three hours at room temperature. Following lysis, puromycin incorporation was measured by western blot with an anti-puromycin antibody (*Figure 5D*). Quantification of three independent experiments revealed no changes in global protein synthesis (*Figure 5E*). We also performed polysome profiling which provides a 'snapshot' of the pool of translationally active ribosomes. Comparison and quantification of profiles obtained from control and *nsun-1* knockdown nematodes did not reveal any differences in the distribution of free subunits, monosomes, and polysomes (*Figure 5F,G*). This agrees with the absence of global protein translation inhibition in the metabolic (puromycin) labeling assay (*Figure 5E*).

In conclusion, NSUN-1 is neither required for pre-rRNA processing nor for global translation. On the contrary, the amounts of ribosomal subunits were reduced in the absence of NSUN-5. The ribosomal biogenesis alterations observed upon *nsun-5* depletion result from a combination of processing inhibitions in ITS2 and increased rRNA intermediates turnover. However, global translation was not detectably affected.

## Loss of *nsun-1* promotes the translation of a distinct subset of mRNAs

Since depletion of *nsun-1* did not affect global protein synthesis, we hypothesized that loss of 26S rRNA m$^5$C methylation might modulate the translation of specific mRNAs, as was previously observed after Rcm1 (NSUN-5 homolog) depletion in yeast (*Schosserer et al., 2015*). To test this possibility, we isolated mRNAs contained in the polysomal fraction, systematically sequenced them by RNA-seq and compared their abundance in polysomes between animals subjected to *nsun-1* RNAi versus RNAi control. Thereby, we identified 52 protein-coding mRNAs to be differentially associated with polysomes between the two conditions (p-adj: 0.05) *Source data 1*. From those, we selected the nine most upregulated and ten most repressed mRNAs based on the magnitude of their fold-change variation (*Figure 6A,B*) and, for comparison, measured their abundance also in polysomes of wild-type (N2) and *nsun-5* knockout (JGG-1) animals by RT-qPCR (*Figure 6C,D*). In contrast to *nsun-1* depletion by RNAi, knockout of *nsun-5* did not significantly affect the presence in polysomes of any of these mRNAs. This suggests that the pattern of translated mRNAs upon *nsun-1* depletion is highly specific and strikingly distinct from that observed upon loss of the other 26S rRNA m$^5$C methyltransferase.

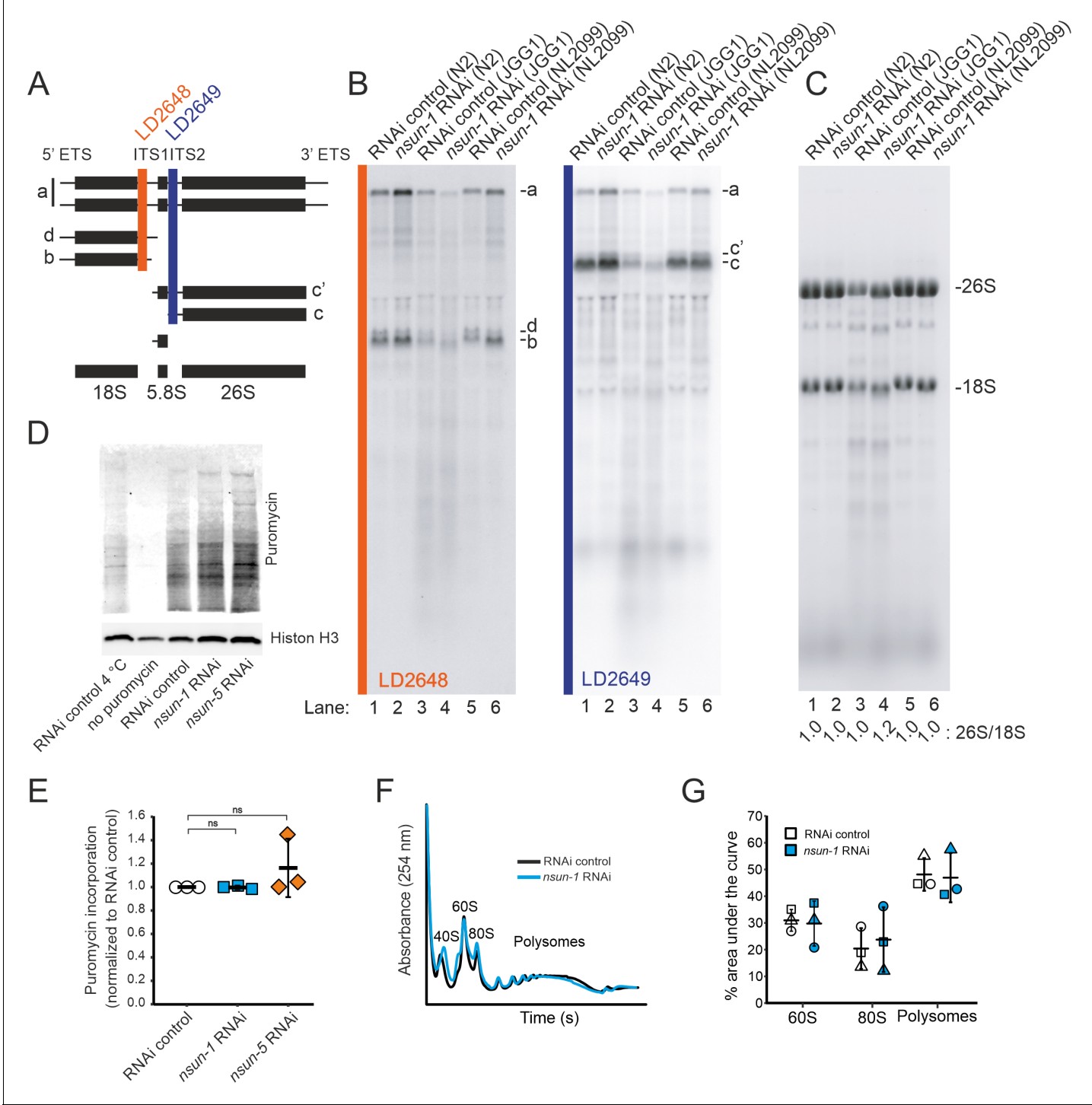

**Figure 5.** NSUN-1 and NSUN-5 are only partially required for rRNA processing and not for global translation. (**A**) Schematics of pre-rRNA processing intermediates in *C. elegans* and probes (LD2648 and LD2649) used in pre-rRNA processing analysis (see panel B). (**B**) Pre-rRNA processing analysis. Total RNA extracted from the indicated strains were separated on denaturing agarose gels and processed for northern blotting. The probes (LD2648 and LD2649) used to detect the pre-rRNA intermediates a, b, c, c´, and d are indicated. (**C**) Steady-state levels of mature rRNAs (18S and 26S) analyzed by ethidium bromide staining and quantified by densitometry. The 26S/18S ratio is indicated. (**D**) Total protein synthesis of N2 animals treated with RNAi control, *nsun-1* or *nsun-5* RNAi. RNAi control treated worms at either 4°C or without puromycin exposure were used as negative controls. Protein synthesis was measured by puromycin exposure for 3 hr and western blot using a puromycin-specific antibody. The experiment was performed in three independent replicates. One representative replicate is shown. Histone H3 was used as loading control. (**E**) Quantification of western blots in D (three biological replicates, one-sample t-test against an expected value of 1, α = 0.05, not significant). (**F–G**) Polysome analysis indicating that global

*Figure 5 continued on next page*

*Figure 5 continued*

translation is not affected by *nsun-1* depletion. Free small subunit (40S), large subunit (60S), monosome (80S) and polysome fractions were detected by UV$_{254}$ monitoring. Representative profiles are shown. (G) Quantification of 60S, 80S, and polysome fractions of three independent experiments reveals no changes between *nsun-1* knockdown and RNAi control.

The online version of this article includes the following source data and figure supplement(s) for figure 5:

**Source data 1.** Raw data of puromycin western blot and polysome profiling quantification.
**Figure supplement 1.** *nsun-5* but not *nsun-1* RNAi alters 5.8S rRNA maturation.

## *nsun-1* knockdown affects GLD-1 expression in the gonad

As changes in polysome abundance upon *nsun-1* depletion can either be caused by transcriptional regulation or by specific recruitment of the respective mRNAs into translating ribosomes, we analyzed total mRNA levels in cell lysates from which then polysome fractionation was performed. We compared total intracellular mRNAs representing the transcriptome, to those contained in the polysomes representing the most actively transcribed mRNAs (in the following designated as 'translatome'). We considered only protein-coding mRNAs with a minimum fold-change of 2 between translatome and transcriptome using an adjusted p-value cut-off at 0.05 (*Figure 7A*; *Source data 2*). We observed that many more mRNAs had their translation repressed (RNAi control: 599, *nsun-1* RNAi: 536) than stimulated (RNAi control: 94, *nsun-1* RNAi: 84). Since the composition of 3' UTRs can affect translation (*Tushev et al., 2018*), we analyzed GC-content, length, and minimal free folding energy of all coding, promoted, and repressed mRNAs in our dataset (*Figure 7B*). Interestingly, all three features significantly differed between RNAi control and *nsun-1* RNAi in promoted and repressed mRNAs (p<0.05), while they remained unchanged when analyzing all coding mRNAs present in our dataset. These findings suggest that loss of *nsun-1* causes the translation of specific subsets of mRNAs based on the composition and length of their 3' UTRs.

Interestingly, 3' UTRs of mRNAs translationally repressed by *nsun-1* depletion were exclusively and significantly enriched (p<0.001) for several binding motifs of ASD-2, GLD-1 and RSP-3 (*Source data 3*). All three RNA-binding proteins are known to play essential roles in *C. elegans* development (*Lee and Schedl, 2010*; *Longman et al., 2000*). Although *gld-1* mRNA was significantly translationally repressed in both *nol-1* RNAi and RNAi control animals by the same magnitude (log2 fold-change: −1.2, p-adj.<0.05, *Source data 2*), neither *asd-2*, *rsp-3* or *gld-1* mRNAs were differentially regulated between *nol-1* RNAi and RNAi control in the transcriptome or in the translatome (*Source data 1*).

GLD-1 is particularly interesting, because levels are highest in the pachytene (also referred to as meiotic zone), where it acts as a translational repressor of mRNAs modulating oogenesis. At the transition zone between the pachytene and the diplotene, GLD-1 levels sharply decrease and previously repressed mRNAs are consequently translated (*Lee and Schedl, 2010*). Since the gonads of *nsun-1* knockdown animals appeared defective precisely at this transition (*Figure 4C*) and GLD-1 target mRNAs were repressed (*Source data 3*), we set out to investigate the effects of *nsun-1* depletion on the spatial distribution of GLD-1 expression during development. For this aim, we used a GLD-1::GFP reporter strain and exposed larvae to RNAi control and *nsun-1* RNAi. Indeed, GLD-1::GFP protein expression was restricted exclusively to a small portion of the loop region in adult nematodes (*Figure 7—figure supplement 1*). Taken together, these findings suggest that *nsun-1* is required for correct gonadal GLD-1 localization during development, which might then directly or indirectly influence the specific translation of mRNAs required for further steps in development based on motifs in their 3' UTRs.

## mRNAs encoding cuticle collagens are translationally repressed upon *nsun-1* knockdown

To further understand the mechanistic link between differential translation and the phenotypes observed upon *nsun-1* knockdown, we performed GO-term enrichment analysis. Among others, GO-terms associated with collagens, structural integrity of the cuticle, and embryo development were significantly enriched amongst the mRNAs which were translationally repressed upon *nsun-1* knockdown (*Figure 7C*, *Source data 4*).

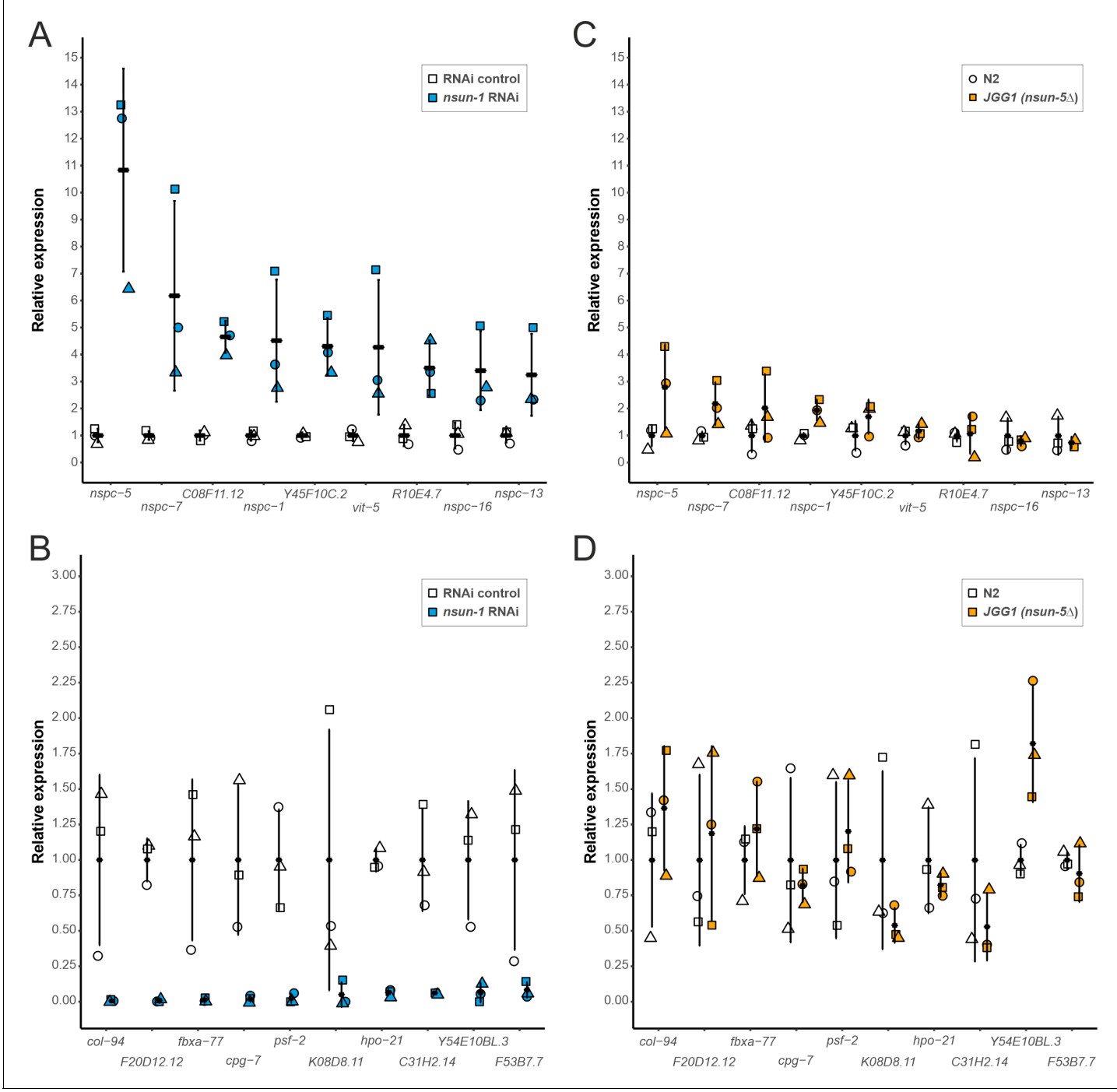

**Figure 6.** Depletion of *nsun-1* modulates the translation of a distinct set of mRNAs which is not altered by *nsun-5* knockout. (**A, B**) mRNA abundance in polysomes of animals subjected to *nsun-1* RNAi and RNAi control was analyzed by RNA-seq. The nine most up- (**A**) and 10 most down-regulated (**B**) protein-coding mRNAs were selected. All shown comparisons between control and *nsun-1* RNAi were statistically significant (adjusted p-value cut-off at 0.05, FDR/Benjamini and Hochberg). (**C, D**) The same mRNAs were quantified in polysomes of wildtype (N2) and *nsun-5* knockout (JGG-1) animals by RT-qPCR. None of the shown comparisons between N2 and JGG1 was statistically significant (adjusted p-value cut-off at 0.05, FDR/Benjamini and Hochberg). Values of each data point were normalized to the mean of the respective control (either RNAi control or N2). Three independent biological replicates were performed. Error bars represent standard deviation.

The online version of this article includes the following source data for figure 6:

**Source data 1.** Raw data of RNA-seq or RT-qPCR of selected polysomal mRNAs upon *nsun-1* RNAi exposure or *nsun-5* knockout.

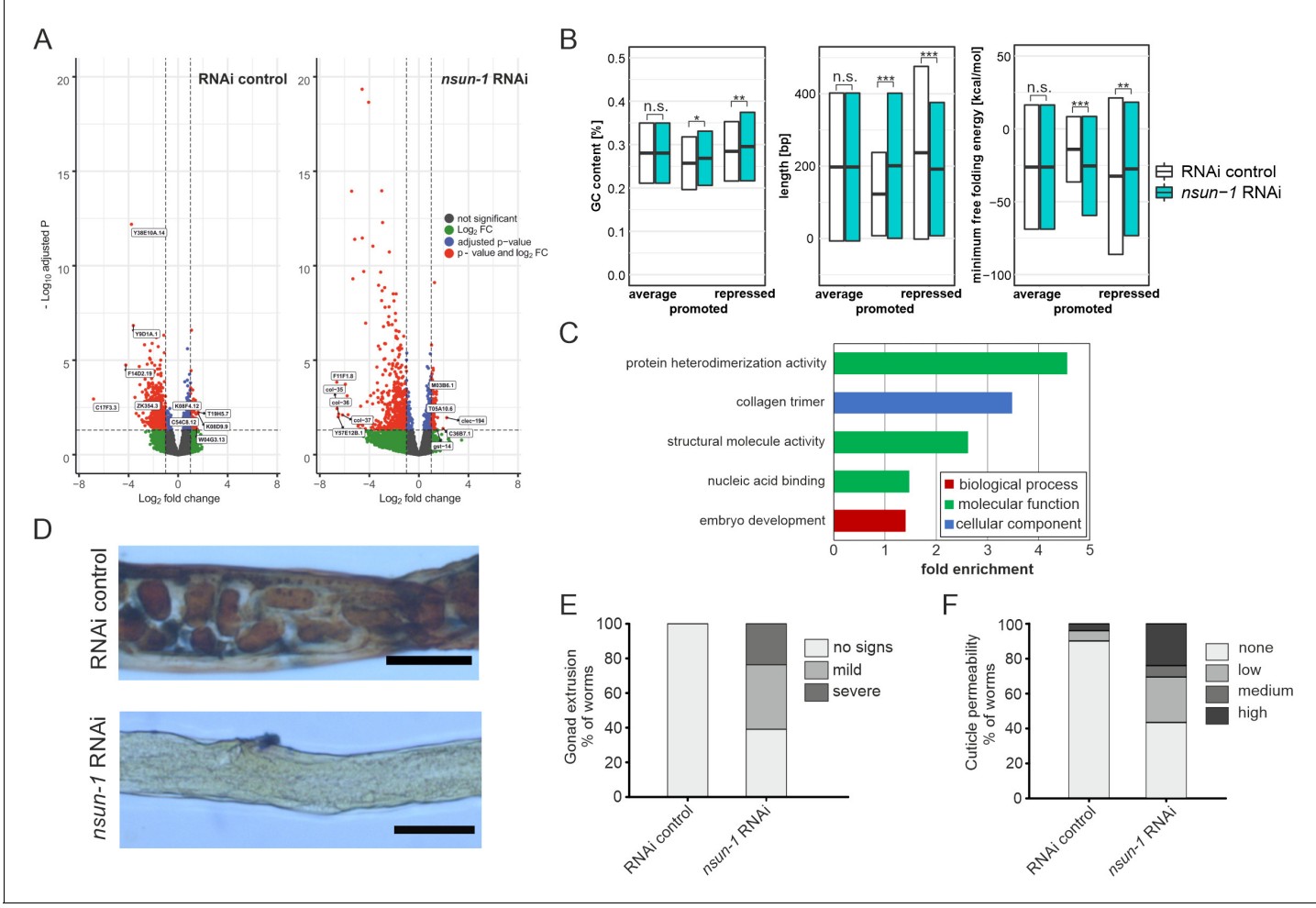

**Figure 7.** *nsun-1* depletion modulates selective translation of collagens and induces gonad extrusion and loss of barrier function. (**A**) Vulcano plots of selectively translated genes after RNAi control and *nsun-1* RNAi exposure. Significantly regulated genes (adjusted p<0.05 and fold-change >2) between polysome fraction and total mRNAs are depicted in red, genes with a two-fold up- or down-regulation but an adjusted p-value (FDR/Benjamini and Hochberg) above 0.05 in green, genes with an adjusted p-value below 0.05 but less than two-fold-change in expression in blue, and not significantly regulated genes in grey. The top five up- or down-regulated genes based on their fold-change are indicated. (**B**) Characteristics of the 3' UTRs of mRNAs with significantly promoted or repressed polysome enrichment (adjusted p<0.1, fold-change >2). GC-content (in %), length (in bp) and minimum free folding energy [kcal/mol] are shown. Boxes indicate mean ± SD (for GC-content and length) or mean ± SEM (for minimum free folding energy). Wilcoxon rank sum test, *p<0.05, **p<0.01, ***p<0.001. (**C**) Biological GO-terms enriched among genes with repressed translation (adjusted p<0.1, fold-change >2) upon *nsun-1* depletion. Modified Fisher's exact test, p<0.05. (**D**) Histological staining (Herovici) to assess collagen deposition. Worms exposed to control RNAi show presence of both young (blue) and mature (pink to brownish-red) collagen whereas animals subjected to *nsun-1* RNAi display less collagen deposition. The cytoplasm is counterstained in yellow. Representative images of the region surrounding the gonad are shown. Two independent experiments with a minimum of 10 animals each were performed with similar outcome. Scale bar, 80 µm. (**E**) Quantification of gonad extrusion upon *nsun-1* depletion compared to RNAi control. 8–9 day old adult animals were classified into three categories according to the severance of gonad extrusion ('no signs', 'mild', 'severe', see *Figure 7—figure supplement 2A*). The experiment was independently performed two times with similar outcome. One representative replicate is shown. n ≥ 50 animals per replicate. Modified Fisher's exact test on the raw count values, p<0.001. (**F**) Quantification of cuticle barrier function upon *nsun-1* depletion compared to RNAi control. Young adult animals were exposed to Hoechst 33342, which is membrane-permeable but cuticle-impermeable. Stained nuclei were counted exclusively in the tail region to exclude intestinal autofluorescence and classified into four categories accordingly ('none', 'low', 'medium', 'high', see *Figure 7—figure supplement 2B*). Three independent experiments were pooled. n(RNAi control)=51, n(*nsun-1* RNAi)=46. Modified Fisher's exact test on the raw count values, p<0.001.

The online version of this article includes the following source data and figure supplement(s) for figure 7:

**Source data 1.** Raw data of brood size, expression during development and body length experiments.
**Figure supplement 1.** GLD-1 localization in the germline is altered by depletion of *nsun-1*.
**Figure supplement 2.** *nsun-1* depletion affects gonad integrity and barrier function.

Since three collagens (*col-35*, *col-36,* and *col-37*) were also found among the five most strongly repressed genes upon loss of *nsun-1* (*Figure 7A*), we decided to assess whether collagen deposition is indeed altered in the animals. In order to test this possibility, we performed a specific histological staining aimed at distinguishing young collagen, detected in blue, from mature collagen, highlighted in pink/brownish-red (*Herovici, 1963*; *Teuscher et al., 2019*). While young adult worms exposed to RNAi control showed presence of both young and mature collagen, animals subjected to *nsun-1* RNAi displayed a strikingly overall reduction of collagen deposition compared to the cytoplasmatic counter-stain (yellow) (*Figure 7D*).

Interestingly, we repeatedly observed an increased fraction of animals displaying gonad extrusion upon *nsun-1* RNAi (see also *Figure 4D*), which might be caused by loss of cuticle structural integrity. To quantify this phenotype precisely, we classified mid-aged animals according to the grade of gonad extrusion which we defined to fall in either of three categories: (i) no visible signs of gonad extrusion, (ii) mild extrusion, or (iii) severe extrusion (*Figure 7—figure supplement 2A*). Upon *nsun-1* depletion, 134 of 220 animals (~60%) showed mild to severe extrusion of the gonad (categories ii and iii), while no extrusion was observed in any of the 50 RNAi control nematodes tested (*Figure 7E*). To further evaluate the possible physiological consequences of altered collagen deposition, we tested cuticle barrier integrity. This assay is based on the principle that the Hochst33342 dye is membrane-permeable, but cuticle-impermeable. As previously described by *Ewald et al., 2015*, worms were grouped into four categories according to whether they were: (i) not permeable (absence of stained nuclei in the animal tail region), (ii) mildly permeable (<5 stained nuclei), (iii) permeable (5–10 stained nuclei), or (iv) highly permeable (>10 stained nuclei) (*Figure 7—figure supplement 2B,C*). Consistent with the reduced production of several collagens, *nsun-1* RNAi caused cuticle permeability (categories ii, iii, and iv) in 26 of 46 animals (~56%) compared to only 5 of 51 RNAi control animals (~10%) (*Figure 7F*).

Taken together, this indicates that NSUN-1 is partially required for translation of several cuticle collagens, which may explain the loss of gonad integrity and the increased cuticle permeability observed upon *nsun-1* depletion. Moreover, several mRNAs whose translation depends on NSUN-1 are associated with embryogenesis.

## Discussion

Although ribosomal RNA modifications are highly conserved in evolution and often present at functionally relevant positions on the ribosomal subunits, only limited information is available on their exact biological functions and, in particular, on their possible involvement in developmental processes or disease etiology (*Sharma and Lafontaine, 2015*). In this work we have investigated the molecular and physiological roles of two structurally related Sun-domain-containing RNA methyltransferases, NSUN-1, and NSUN-5 in *C. elegans*. Each enzyme is responsible for writing one specific m$^5$C mark on 26S rRNA. We further describe NSUN-1 as a *bona fide* m$^5$C rRNA writer enzyme that, if missing, directly entails physiological and developmental consequences. We conclude that, molecularly, loss of NSUN-1 function leads to translational remodeling with profound consequences on cell homeostasis, exemplified by loss of cuticle barrier function, and highly specific developmental defects, including oocyte maturation failure. We further suggest that extrusion of the gonad and loss of cuticle barrier function are directly caused by reduced expression of collagens, while the developmental defects are associated with altered expression and/or localization of several important developmental regulators such as GLD-1. We summarized the observed RNAi phenotypes in different worm strain backgrounds in *Table 2*.

According to the 'disposable soma theory of aging', a balance between somatic repair and reproduction exists. Depending on its environment, an organism may direct the available energy either to maintenance of the germline thereby ensuring efficient reproduction, or to the homeostasis of somatic cells including the prevention of DNA damage accumulation (*Kirkwood and Holliday, 1979*). Accordingly, most of the known genetic or nutritional interventions that increase the lifespan of organisms antagonistically reduce growth, fecundity, and body size (*Kapahi, 2010*; *Kenyon et al., 1993*). Indeed, reduction of overall protein synthesis by genetic, pharmacological, or dietary interventions was reproducibly shown to extend longevity in different aging model organisms (*Chiocchetti et al., 2007*; *Curran and Ruvkun, 2007*; *Hansen et al., 2007*; *Kaeberlein et al., 2005*; *Masoro, 2005*; *Pan et al., 2007*). These reports clearly established protein synthesis as an important

**Table 2.** Comparison of phenotypes after *nsun-1* and *nsun-5* depletion, n.d.: not determined.

| Phenotype | nsun-1 RNAi | nsun-5 RNAi |
|---|---|---|
| Lifespan | - Unaffected in whole adult treatment<br>- Unaffected after germline-specific depletion<br>- Increased by ~ 10% after soma-specific depletion | - Increased by ~ 17% in whole adult treatment (*Schosserer et al., 2015*) |
| Stress resistance (heat) in adults | Increased | Similarly increased |
| Locomotion at midlife | Increased | Similarly increased |
| Brood size (fecundity) | Reduced (2-fold) | Unaffected |
| Adult animal size | - Reduced by ~ 20% after soma-specific depletion and whole-body depletion<br>- Unaffected after germline-specific depletion | Unaffected |
| Gonad morphology | - Impaired at meiotic to oocyte transition<br>-Gonad extrusion (possibly caused by loss of cuticle integrity)<br>- Unaffected after germline-specific depletion | Unaffected |
| Pre-rRNA processing | Unaffected | Affected |
| Collagen expression | Affected (translational remodeling) | n.d. |
| Cuticle permeability | increased | n.d. |

regulator of the aging process at the interface between somatic maintenance and reproduction. Thus, we were surprised to find that despite their ability to modulate aging and to methylate rRNA, neither NSUN-1 nor NSUN-5 were required for global protein synthesis in worms under the conditions tested. In the case of NSUN-5 depletion, we previously found overall translation to be decreased in mammalian cells (*Heissenberger et al., 2019*), but not in yeast (*Schosserer et al., 2015*). We reasoned that the higher complexity of mammalian ribosomes and associated factors might render them more vulnerable to alterations of rRNA secondary structure, for example caused by loss of a single base modification, than ribosomes from yeast or nematodes.

As ribosome biogenesis and global translational activity per se were not severely affected by loss of NSUN-1, the principle that specialized ribosomes modified at specific positions may be particularly efficient to translate selectively mRNAs important for worm development and physiology was appealing to us (*Simsek and Barna, 2017*). Indeed, lack of NSUN-1 and thus of methylation at C2982 resulted in decreased translation of mRNAs containing GLD-1 and ASD-2 binding sites. These two proteins are closely related members of the STAR protein family involved in mRNA-binding, splicing and nuclear export of mRNAs. While the molecular functions of ASD-2 are only poorly understood, the role of GLD-1 in embryonic development is well characterized (*Lee and Schedl, 2010*). Since the gonads of *nsun-1* knockdown animals appeared defective precisely at the transition between the pachytene and the diplotene and GLD-1 target mRNAs were repressed, we speculate that either ribosomes lacking the methylation at C2982 have generally low affinity for these mRNAs, or that translational repression by GLD-1 is never fully relieved. Although the expression of GLD-1 itself was not differentially regulated between control and *nsun-1* depleted animals at the transcriptional or translational levels (*Source data 1*), multiple direct or indirect connections to NSUN-1 which modulates ribosome function are still conceivable and will require further studies.

Previously, Curran and Ruvkun reported that depletion of *nsun-1* (W07E6.1) by adult-onset RNAi treatment led to extended lifespan in *C. elegans* (*Curran and Ruvkun, 2007*). In their high throughput screen the authors used a strain carrying a mutation in the *eri-1* gene, rendering it hypersensitive to RNAi in the whole body, but especially in neurons and in the somatic gonad (*Kennedy et al., 2004*). In this study, we conducted a whole-body knockdown in N2 wild-type animals but could not confirm these previously observed effects on lifespan, although the health status of mid-aged nematodes, as assessed by quantifying locomotion behavior, were slightly improved. However, when knocking-down *nsun-1* specifically in somatic tissues, but not in the germline, we observed a clear lifespan extension. The N2 wild-type strain is usually resistant to RNAi in the somatic gonad and neurons. Thus, we hypothesize that depletion of *nsun-1* specifically in the somatic part of the gonad is required for lifespan extension, which is only effectively realized in the *eri-1* and *ppw-1* mutant strains, but not in N2 wild-type animals. Intriguingly, these findings further suggest possible non-

cell-autonomous effects of single RNA methylations, since modulation of NSUN-1 levels in somatic cells profoundly affected distinct cells of the germline.

The developing gonad of L1 larvae consists of two primordial germ cells and two surrounding somatic gonad precursor niche cells. The crosstalk between these two cell types, which form the germline and somatic part of the gonad at later stages of larval development, was already described to modulate aging and stress responses. Laser depletion of both primordial germ cells extends lifespan via insulin/IGF-signaling, while animals with an additional depletion of the two somatic gonad precursor cells have a normal lifespan (*Hsin and Kenyon, 1999*). Of potential relevance to our study is a recent report by Ou and coworkers, who demonstrated that IFE-4 regulates the response to DNA damage in primordial germ cells in a non-cell-autonomous manner via FGF-like signaling. Soma-specific IFE-4 is involved in the specific translation of a subset of mRNA including *egl-15*. Thereby, IFE-4 regulates the activity of CEP-1/p53 in primordial germ cells despite not being present there (*Ou et al., 2019*). We thus hypothesize that selective translation of mRNAs by specialized ribosomes, either generated by association with translational regulators such as IFE-4, or by RNA modifications as described here, might serve as a general mechanism to tightly control essential cellular processes even in distinct cells and tissues.

Accumulating data suggests that NSUN-1 has pleiotropic effects independent of its methylation activity. While knockout of *nsun-1* appears to be lethal (*Figure 1—figure supplement 1*), a catalytic mutant strain of *nsun-1* only lacking its methylation activity was still viable (*Navarro et al., 2020*). Similarly, deletion of Nop2, the yeast homolog of *nsun-1*, was shown to be lethal, but viability could be restored by re-expression of a catalytic mutant (*Sharma et al., 2013*).

Elucidating the precise mechanisms of NSUN1 function is of prime importance, as the human protein (also known as NOP2 or P120) was shown to be required for mammalian preimplantation development (*Cui et al., 2016*), which, when considering the effects we report on gonad maturation in worm, highlights evolutionary conservation. Cui and colleagues found that NSUN1 carries an essential role during blastocyst development within their experimental system. Interestingly, NSUN1 appears to be important in other physiological contexts as well, as it was recently shown to restrict HIV-1 replication and promote latency by participating in the methylation of the HIV-1 TAR RNA (*Kong et al., 2020*). Additionally, other groups reported that low levels of NSUN1 reduce cell growth in leukemia cells, which is in line with the findings that NSUN1 promotes cell proliferation. Moreover, high NSUN1 expression results in increased tumor aggressiveness and augmented 5-azacytidine (5-AZA) resistance in two leukaemia cell lines (*Bantis et al., 2004*; *Cheng et al., 2018*; *Saijo et al., 2001*). Thus, NSUN1 might be considered as an example of 'antagonistic pleiotropy'. According to this theory, genes can be essential early in life and become dispensable later, for instance after sexual reproduction. While NSUN1 appears to be essential for normal development, it might increase tumor aggressiveness later in life, especially in highly proliferative cells and tissues.

## Materials and methods

**Key resources table**

| Reagent type (species) or resource | Designation | Source or reference | Identifiers | Additional information |
|---|---|---|---|---|
| gene (*Caenorhabditis elegans*) | *nsun-1* | WormBase | WBGene00021073 | Also known as: *nol-1*, *nol-2* |
| gene (*C. elegans*) | *nsun-5* | WormBase | WBGene00013151 | |
| strain, strain background (*C. elegans*) | N2 | CGC, University of Minnesota | RRID:WB-STRAIN:WB Strain00000001 | Genotype: wildtype |
| genetic reagent (*C. elegans*) | FX30263 | National Bioresource Project, Tokyo, Shohei Mitani | | Genotype: *nsun-1(tm6081) II/lin-42(tmIs1246) II* |

*Continued on next page*

*Continued*

| Reagent type (species) or resource | Designation | Source or reference | Identifiers | Additional information |
|---|---|---|---|---|
| genetic reagent (*C. elegans*) | JGG1 | CGC, University of Minnesota | RRID:WB-STRAIN:WB Strain00022241 | Genotype: *nsun-5(tm3898) II* |
| genetic reagent (*C. elegans*) | SA115 | CGC, University of Minnesota | RRID:WB-STRAIN:WB Strain00033882 | Genotype: *unc-119(ed3) III; tjIs1 [pie-1::GFP:: rho-1 + unc-119(+)]* |
| genetic reagent (*C. elegans*) | JJ1473 | CGC, University of Minnesota | RRID:WB-STRAIN:WB Strain00022491 | Genotype: *unc-119(ed3) III; zuIs45 [nmy-2p::nmy-2::GFP + unc-119(+)] V* |
| genetic reagent (*C. elegans*) | TP12 | CGC, University of Minnesota | RRID:WB-STRAIN:WB Strain00034928 | Genotype: *kaIs12[col-19::GFP]* |
| genetic reagent (*C. elegans*) | DCL569 | CGC, University of Minnesota | RRID:WB-STRAIN:WB Strain00005607 | Genotype: *mkcSi13 [sun-1p::rde-1::sun-1 3'UTR + unc-119(+)] II* |
| genetic reagent (*C. elegans*) | NL2098 | CGC, University of Minnesota | RRID:WB-STRAIN:WB Strain00028994 | Genotype: *rrf-1(pk1417) I* |
| genetic reagent (*C. elegans*) | NL2550 | CGC, University of Minnesota | RRID:WB-STRAIN:WB Strain00029002 | Genotype: *ppw-1(pk2505) I* |
| genetic reagent (*C. elegans*) | JK4626 | CGC, University of Minnesota | RRID:WB-STRAIN:WB Strain00022650 | Genotype: *cku-80(ok861) unc-119(ed3) III; qIs170 [gld-1p::gld-1::GFP::FLAG + unc-119(+)]* |
| antibody | Anti-puromycin antibody, mouse monoclonal | Millipore | Cat# MABE343, RRID:AB_2566826 | Western Blot: (1:10000) |
| antibody | Anti-Histone H3 antibody, rabbit polyclonal | Abcam | Cat# ab1791, RRID:AB_302613 | Western Blot: (1:4000) |
| antibody | Anti-Rabbit-IR-Dye 800, donkey polyclonal | LI-COR Biosciences | Cat# 926–32213, RRID:AB_621848 | Western Blot: (1:10000) |
| antibody | Anti-Mouse-IR-Dye 680RD, donkey polyclonal | LI-COR Biosciences | Cat# 926–68072, RRID:AB_10953628 | Western Blot: (1:10000) |
| recombinant DNA reagent | RNAi control (empty vector) | Addgene | RRID:Addgene_1654 | Vector: L4440 Host Strain: HT115 (DE3) |
| recombinant DNA reagent | RNAi clone (*nsun-1*) | Source Bioscience | Cat# CUUkp 3301A161Q | Vector: L4440 Host Strain: HT115 (DE3) |
| recombinant DNA reagent | RNAi clone (*nsun-5*) | *Schosserer et al., 2015* PMID:25635753 | | Vector: L4440 Host Strain: HT115 (DE3) |
| sequence-based reagent | PCR primers | This paper | | See *Supplementary file 1* |
| sequence-based reagent | LD2648 (ITS1) | *Bar et al., 2016* PMID:27457958 | | Northern blot probe, sequence: CACTCAACTGACCG TGAAGCCAGTCG |
| sequence-based reagent | LD2649 (ITS2) | *Bar et al., 2016* PMID:27457958 | | Northern blot probe, sequence: GGACAAGATCAGT ATGCCGAGACGCG |

*Continued on next page*

*Continued*

| Reagent type (species) or resource | Designation | Source or reference | Identifiers | Additional information |
|---|---|---|---|---|
| commercial assay or kit | Direct-zol RNA Miniprep | Zymo Research | Cat# R2051 | |
| commercial assay or kit | EZ RNA Methylation Kit | Zymo Research | Cat# R5001 | |
| commercial assay or kit | ExACT Genotyping Kit | BioCat | Cat# 2212–500-BL | |
| commercial assay or kit | High-Capacity cDNA Reverse Transcription Kit | Life Technologies | Cat# 4368814 | |
| commercial assay or kit | 5x HOT FIREPol EvaGreen qPCR Mix | Medibena | Cat# SB_08–24-GP | |
| chemical compound, drug | 5-Fluoro-2′-deoxyuridine (FUdR) | Sigma Aldrich | Cat# F0503-100MG | |
| chemical compound, drug | Puromycin | Invivogen | Cat# ant-pr-1 | |
| chemical compound, drug | TRIzol LS Reagent | Life Technologies | Cat# 10296028 | |
| software, algorithm | Image J | Image J | Fiji, RRID:SCR_002285 | Version 2.0.0-rc-65/1.51 w; Java 1.8.0_162 [64-bit] |
| software, algorithm | WormLab | MBF Bioscience | | Version 4.1.1 |
| software, algorithm | R | The R Foundation for Statistical Computing | | Version 4.0.3 Script for RNA-seq analysis: *Source code 1* |
| software, algorithm | SigmaPlot | Systat Software Inc | | Version 14 |
| software, algorithm | Prism | GraphPad | | Version 9.0.0 |
| software, algorithm | Galaxy | Galaxy Project | RRID:SCR_006281 | https://usegalaxy.org/ Version numbers of individual tools are indicated in Materials and methods |

## Worm strains and culture conditions

Worm strains used in this study are listed in the Key Resources Table. Worms were cultured following standard protocols on *Escherichia coli* OP50-seeded NGM-agar plates at 20℃, unless indicated otherwise (*Brenner, 1974*).

## Genotyping of *tm6081*

Genotyping was performed using the ExACT Genotyping Kit (BioCat). Single nematodes were picked from plates into 5 µL nuclease-free water in PCR tubes, taking care to minimize bacterial contamination. 1 µL Buffer A and 0.5 µL Buffer B were added to each tube and incubated at 75℃ for 5 min and for 10 min at 95℃ in a thermocycler with heated lid. Then, PCR was performed following the manufacture's protocol using specific primers spanning the *tm6081* allele (*Supplementary file 1*) in 25 µL reaction volume, 50℃ annealing temperature and 35 cycles.

## RNAi knockdown

For inactivating *nsun-1* and *nsun-5*, feeding of double-stranded RNA expressed in bacteria was used (*Timmons et al., 2001*). Therefore, the HT115 strain of *E. coli*, carrying either the respective RNAi construct or the empty vector (L4440) as RNAi control, was cultured overnight in LB medium with ampicillin and tetracyclin at 37°C. Bacteria were harvested by centrifugation, resuspended in LB medium and either 100 µL (60 mm plates) or 400 µL (100 mm plates) were plated on NGM containing 1 mM isopropyl-b-D-thiogalactoside and 25 µg/mL carbenicillin. The plates were incubated at 37°C overnight and used within one week.

Larval-onset RNAi was achieved by bleaching adult animals. Released eggs were transferred directly to plates seeded with RNAi bacteria. Adulthood was usually reached after three days and animals were used for experiments when the RNAi control strain started to lay eggs.

In case of adult-onset RNAi, eggs were transferred to plates seeded with RNAi control bacteria. Animals were raised until egg production commenced and subsequently transferred to the respective RNAi bacteria.

## Differential Interference Contrast (DIC) and fluorescence microscopy

Worms were paralyzed using 1 M sodium azide solution and mounted on 2% agar pads. Images were acquired on a Leica DMI6000B microscope with a 10x dry objective (NA 0.3), a 20x dry objective (NA 0.7), or a 63x glycerol objective (NA 1.3) in DIC brightfield or fluorescence mode. Confocal microscopy presented in *Figure 4C* was performed on a Leica TCS SP5 spot scanning confocal microscope equipped with an HCX PL APO CS 40x/0.85 dry objective, HyD detector and Argon-laser. Cropping, insertion of scale bars and brightness and contrast adjustments were done with Image J (version 2.0.0-rc-65/1.51 w; Java 1.8.0_162 [64-bit]).

## Mobility

Animals were either synchronized by timed egg-lay (two replicates) or by hypochlorite treatment (one replicate) on RNAi control plates. When reaching adulthood, nematodes were transferred to RNAi plates. Every few days at regular intervals, plates were rocked in order to induce movement of animals and videos were subsequently recorded for one minute. Worms were transferred to fresh plates whenever necessary. At day 16 the vast majority of worms completely ceased movement, thus we did not include any later timepoints. Notably, we did not notice any obvious aversion behavior or elevated speed at young age upon *nsun-1* or *nsun-5* RNAi, which was previously shown to be present upon depletion of other components of the translational machinery (*Melo and Ruvkun, 2012*). Worm Lab version 4.1.1 was used to track individual animals and calculate the average speed.

## Lifespan assays

Lifespan measurement was conducted as previously described (*Schosserer et al., 2015*). For lifespan assays, 90 adults per condition were transferred to plates seeded with the respective RNAi bacteria (control, *nsun-1*, *nsun-5*). Wildtype worms were pre-synchronized on NGM plates seeded with UV-killed OP50 bacteria. 50 adult worms were transferred to NGM plates and allowed to lay eggs for 15 hr; then the adult worms were removed. Synchronization by timed egg-lay was performed 72 hr after the pre-synchronization by transferring 350 gravid worms from the pre-synchronization to fresh NGM plates seeded with RNAi control bacteria and allowed to lay eggs for four hours. After 68 hr, 90 young adult worms per condition were placed on fresh NGM plates containing 5 mL NGM, 100 µL bacterial suspension and 50 µg FUdR. This day represents day 0 in the lifespan measurement. Worms were scored as 'censored' or 'dead' every two to four days. Nematodes were scored as 'censored' if they had crawled off the plate, were missing or died due to other causes than aging, such as gonad extrusion. Animals were transferred to fresh plates every 3–7 days depending on the availability of the bacterial food source. Lifespans were performed at 20°C. Kaplan-Meier survival curves were plotted and log-rank statistics were calculated.

## Thermotolerance

Thermotolerance was assessed as previously described (*Vieira et al., 2018*). Animals were synchronized by hypochlorite treatment and released eggs were transferred to NGM plates seeded with RNAi control bacteria and kept at 20°C. After 48 hr, L4 animals were picked on RNAi control, *nsun-1*

or *nsun-5* RNAi plates and exposed to RNAi for approximately three days (68 hr). Subsequently, plates were transferred to 35℃ and scored every 1–2 hr for survival.

## Body size

Worms were synchronized by hypochlorite treatment and incubated in liquid S-Basal medium overnight. On the following day, eggs/L1 were transferred to RNAi plates (RNAi control, *nsun-1,* and *nsun-5* RNAi). Three days later, worms were transferred to agar pads and paralyzed using sodium azide and visualized using DIC microscopy (see above).

## Brood size analysis

Worms were synchronized by treatment with hypochlorite solution and incubated in S-Basal at room temperature overnight. L1 larvae were subsequently transferred to NGM plates seeded with RNAi control bacteria. After 48 hr, L4 animals were transferred to individual wells of a 24-well plate seeded with the respective RNAi bacteria (HT115, *nsun-1*, *nsun-5*). Each well contained 1.5 mL of NGM-agar and 3 µL of bacterial suspension (1:2 dilution in S-Basal). Worms were transferred to a new well every day for four consecutive days and total progeny of individual animals was counted. Per condition and experiment, five worms were analyzed.

## Global protein synthesis by puromycin incorporation

Puromycin incorporation was measured as previously described (*Tiku et al., 2018*) with minor modifications. Heat-inactivated OP-50 (75℃, 40 min) were provided as food source during pulse-labeling. As negative controls, RNAi control treated worms were used either without addition of puromycin or by pulse-labeling at 4℃ instead of 20℃. Around 100 animals per condition were harvested for western blot analysis. Lysis was done directly in SDS loading dye (60 µM Tris/HCl pH 6.8, 2% SDS, 10% glycerol, 0.0125% bromophenol blue and 1.25% β-mercaptoethanol). Worms in SDS loading dye were homogenized with a pellet pestle for 1 min. Then, the samples were heated to 95℃ and loaded on 4–15% Mini-PROTEAN TGX gels (BioRad) in Laemmli-Buffer (25 mM Tris, 250 mM glycine and 0.1% SDS). Protein bands were transferred to PVDF-membranes (Bio Rad) at 25 V and 1.3 A for 3 min. After blocking with 3% milk in PBS, the membrane was incubated overnight at 4℃ with a mixture of anti-Histone H3 (Abcam ab1791, 1:4000) and anti-puromycin (Millipore 12D10, 1:10000). After washing and secondary antibody incubation (IRDye680RD and IRDye800CW, 1:10000), the membrane was scanned on the Odyssey Infrared Imager (LI-COR). Quantification of band intensities was performed in Image J (version 2.0.0-rc-65/1.51 w; Java 1.8.0_162 [64-bit]).

## Polysome profiling

Two-day-old adult worms were used to generate polysome profiles as previously described (*Rogers et al., 2011*). One hundred microliter worm-pellet were homogenized on ice in 300 µL of solubilization buffer (300 mM NaCl, 50 mM Tris-HCl (pH 8.0), 10 mM $MgCl_2$, 1 mM EGTA, 200 µg/mL heparin, 400 U/mL RNAsin, 1.0 mM phenylmethylsulfonyl fluoride, 0.2 mg/mL cycloheximide, 1% Triton X-100, 0.1% sodium deoxycholate) using a pellet pestle. 700 µL additional solubilization buffer were added, vortexed briefly, and placed on ice for 10 min before centrifugation at 20.000 g for 15 min at 4℃. Approximately 0.9 mL of the supernatant was applied to the top of a linear 10–50% sucrose gradient in high salt resolving buffer (140 mM NaCl, 25 mM Tris-HCl (pH 8.0), 10 mM $MgCl_2$) and centrifuged in a Beckman SW41Ti rotor (Beckman Coulter, Fullerton, CA, USA) at 180.000 g for 90 min at 4℃. Gradients were fractionated while continuously monitoring the absorbance at 260 nm.

## RNA-seq

Trizol LS (Life Technologies) was immediately added to collected fractions and RNA was isolated following the manufacturer's protocol. PolyA-selection, generation of a strand-specific cDNA library and sequencing on the HiSeq 4000 platform (Illumina) using the 50 bp SR mode was performed by GATC Biotech (Konstanz, Germany). At least 30 million reads were generated per sample.

FASTQ Trimmer by column (Galaxy Version 1.0.0) was used to remove the first 12 bases from the 5' end of each read due to an obvious base bias in this region, as detected by FastQC (Galaxy Version 0.69). Filter by quality (Galaxy Version 1.0.0) was performed using a cut-off value of 20 and only

reads with a maximum number of 8 bases with quality lower than the cut-off value were retained. RNA STAR (Galaxy Version 2.6.0b-1) was used to align reads to the WBcel235 reference genome using the default options. Aligned reads with a minimum alignment quality of 10 were counted using htseq-count (Galaxy Version 0.9.1).

Differential expression was analyzed using the DEseq2 package in R. The contrast
~batch + condition (batch = biological replicate, condition = sample description)
was applied to compare the polysome fraction to the total RNA of either RNAi control or *nsun-1* RNAi-treated samples. Afterwards, results were filtered in R to contain only protein-coding genes (according to ENSEMBL annotation), genes with detectable expression (base mean >1), a fold-change of >2 (log2FC > 1) and an adjusted p-value of <0.05. Volcano plots were generated in R using the Enhanced Volcano package, labeling the top five up- and down-regulated genes respectively.

GO-term enrichment using DAVID (version 6.7) was performed as previously described (*Rollins et al., 2019*). Only protein-coding genes with detectable expression (base mean >1), a fold-change of >2 and an adjusted p-value of <0.10 were considered. For visualization, only the broadest GO-terms of the GOTERM_BP_FAT, GOTERM_MF_FAT and GOTERM_CC_FAT categories, which were still significantly enriched (FDR < 0.05), are shown while similar terms based on the same subset of genes but lower in hierarchy were manually removed. Full results are contained in the supplements.

UTR characterization and RBP motif enrichment were performed as previously described (*Rollins et al., 2019*) using only protein-coding genes with detectable expression (base mean >1), a fold-change of >2 and an adjusted p-value of <0.10.

The raw and processed sequencing data are available from the Gene Expression Omnibus database (https://www.ncbi.nlm.nih.gov/geo) under accession GSE143618.

The R-script for analyzing RNA-seq data is provided as *Source code 1*.

## RT-qPCR

All primer sequences are provided in *Supplementary file 1*. Samples were collected by either transferring worms individually into 1.5 mL tubes, by washing them off NGM plates using S-Basal or from polysome fractions and lysates (see above). After three washing steps with S-Basal, 300 µL TRIzol LS Reagent were added to approximately 100 µL residual S-Basal including worms. Subsequently, worms were homogenized with a pellet pestle for one minute, 600 µL TRIzol LS Reagent were added and the sample was vortexed for five minutes at room temperature. Total RNA was isolated using Direct-zol RNA MiniPrep Kit (Zymo) according to the instructions by the manufacturer. For cDNA synthesis, 500 ng RNA were converted into cDNA using the Applied Biosystems High-Capacity cDNA Reverse Transcription Kit (Thermo Fisher Scientific). cDNA was amplified from total RNA using random primers. RT-qPCR was performed on a Rotor-Gene Q (QIAGEN) using HOT FIREPol EvaGreen qPCR Mix. The absolute amounts of mRNAs were calculated by computing a standard curve and the resulting copy numbers were normalized to the housekeeping genes *act-1* and *tba-1*.

For measuring mRNA expression during development, worms were synchronized by treatment with hypochlorite solution and the released eggs were subsequently transferred to four separate NGM plates seeded with UV-killed OP50 bacteria. Samples were taken from eggs immediately after bleaching, L1/L2 (20 hr after bleaching), L3 (32 hr after bleaching), L4 (46 hr after bleaching) and young adults (60 hr after bleaching).

## 3-D ribosome structure

The PyMOL Molecular Graphics System (Version 2.0) was used. The structure was modeled on the human 80S ribosome (PDB 6EK0).

## m⁵C detection by COBRA assay

NSUN-5 activity was measured by the COBRA assay as previously described (*Adamla et al., 2019*). Primer sequences are provided in *Supplementary file 1*.

## HPLC analysis of m$^5$C

13–15 µg 26S purified on sucrose gradient were digested to nucleosides and analyzed by HPLC. Peaks elutes at 12 min and as a control, a commercial 5-methylcytidine (NM03720, CarboSynth) was used. For quantification of m$^5$C peak area, the peak was normalized to either the peak eluting at 16 min (asterisk on the Figure), or to the peak eluting at 8 min (U), with similar results. The results are shown for normalization to the peak eluting at 16 min.

## Pre-rRNA processing analysis

For analysis of high–molecular weight RNA species, 3 µg total RNA was resolved on a denaturing agarose gel (6% formaldehyde/1.2% agarose) and migrated for 16 hr at 65 volts. Agarose gels were transferred by capillarity onto Hybond-N+ membranes. The membrane was prehybridized for 1 hr at 65°C in 50% formamide, 5x SSPE, 5x Denhardt's solution, 1% SDS (w/v) and 200 µg/mL fish sperm DNA solution (Roche). The $^{32}$P-labeled oligonucleotide probe (LD2648 (ITS1): CACTCAACTGACCG TGAAGCCAGTCG; LD2649 (ITS2): GGACAAGATCAGTATGCCGAGACGCG) was added and incubated for 1 hr at 65°C and then overnight at 37°C. For analysis of low molecular weight RNA species, northern blots were exposed to Fuji imaging plates (Fujifilm) and signals acquired with a Phosphorimager (FLA-7000; Fujifilm).

## Statistics and sample size estimation

No explicit power analysis was used. Sample sizes estimations were partially based on our own previous empirical experience with the respective assays, as well as the cited literature.

No systematic blinding of group allocation was used, but samples were always analyzed in a random order. Nematodes were randomly assigned to the experimental groups. All lifespan, stress resistance and locomotion experiments were performed by at least two different operators.

Most experiments were performed in three independent experiments, unless stated otherwise in the figure legend. Independent experiments were always initiated at different days and thus always resemble different batches of nematodes. Some experiments (RNA isolation for RNA-seq, pre-rRNA processing analysis) were performed once with all frozen independent batches of nematodes to minimize technical variation. No outliers were detected or removed. Criteria for censoring animals for lifespan, stress resistance, and locomotion experiments are indicated in the respective chapters.

Statistical tests used, exact values of N, definitions of center, methods of multiple test correction, and dispersion and precision measures are indicated in the respective figure legends. P-value thresholds were defined as *p<0.05, **p<0.01, and ***p<0.001. For RNA-seq, statistical tests and p-value thresholds are explained in detail in the 'RNA-seq' chapter.

## Acknowledgements

We are grateful to Tamás Barnabás Könye for technical assistance and the BOKU Core Facilities Multiscale Imaging for technical support with microscopy. This work was supported by the Austrian Science Fund (FWF) and Herzfelder'sche Familienstiftung [P30623 to MS], Hochschuljubiläumsstiftung der Stadt Wien [H- 327123/2018 to MS], and the Austrian Science Fund (FWF) [I2514 to JG]. Research reported in this publication was supported by the James L Boyer Fellowship at the MDI Biological Laboratory to MS. Research conducted in the labs of ANR and JAR was supported by an Institutional Development Award (IDeA) from the National Institute of General Medical Sciences of the National Institutes of Health under grant numbers P20GM103423 and P20GM104318. Research in the Lab of DLJL is supported by the Belgian Fonds de la Recherche Scientifique (FRS/FNRS), the Université Libre de Bruxelles (ULB), the Région Wallonne (DGO6) [grant RIBO*cancer* n°1810070], the Fonds Jean Brachet, the International Brachet Stiftung, and the Epitran COST action (CA16120). FN is a fellow of the international PhD programme 'BioToP-Biomolecular Technology of Proteins', funded by the Austrian Science Fund (FWF) [W1224 to JG]. Some strains were provided by the CGC, which is funded by NIH Office of Research Infrastructure Programs (P40 OD010440). We gratefully acknowledge Prof. Shohei Mitani for providing us with strains through the National Bioresource Project.

## Additional information

### Funding

| Funder | Grant reference number | Author |
|--------|------------------------|--------|
| Austrian Science Fund | P30623 | Markus Schosserer |
| Herzfelder'sche Familienstiftung | P30623 | Markus Schosserer |
| Hochschuljubiläumsstiftung der Stadt Wien | H- 327123/2018 | Markus Schosserer |
| Austrian Science Fund | I2514 | Johannes Grillari |
| National Institute of General Medical Sciences | P20GM103423 | Jarod A Rollins Aric N Rogers |
| Université Libre de Bruxelles (ULB) | | Denis L J Lafontaine |
| Mount Desert Island Biological Laboratory | James L. Boyer Fellowship | Markus Schosserer |
| National Institute of General Medical Sciences | P20GM104318 | Jarod A Rollins Aric N Rogers |
| Belgian National Fund for Scientific Research | F.R.S./FNRS | Denis L J Lafontaine |
| Région Wallonne (DGO6) | grant RIBO cancer no. 1810070 | Denis L J Lafontaine |
| Austrian Science Fund | W1224 | Johannes Grillari |

The funders had no role in study design, data collection and interpretation, or the decision to submit the work for publication.

### Author contributions

Clemens Heissenberger, Conceptualization, Formal analysis, Investigation, Visualization, Writing - original draft; Jarod A Rollins, Resources, Software, Formal analysis, Funding acquisition, Investigation, Visualization, Methodology, Writing - review and editing; Teresa L Krammer, Formal analysis, Investigation, Visualization; Fabian Nagelreiter, Software, Formal analysis, Visualization, Writing - review and editing; Isabella Stocker, Ludivine Wacheul, Anton Shpylovyi, Formal analysis, Investigation; Koray Tav, Santina Snow, Investigation; Johannes Grillari, Resources, Supervision, Funding acquisition, Writing - review and editing; Aric N Rogers, Resources, Supervision, Funding acquisition, Methodology, Writing - review and editing; Denis L J Lafontaine, Formal analysis, Supervision, Funding acquisition, Methodology, Writing - review and editing; Markus Schosserer, Conceptualization, Software, Formal analysis, Supervision, Funding acquisition, Validation, Investigation, Visualization, Methodology, Writing - original draft, Project administration

### Author ORCIDs

Denis L J Lafontaine (ID) https://orcid.org/0000-0001-7295-6288
Markus Schosserer (ID) https://orcid.org/0000-0003-2025-0739

### Decision letter and Author response

Decision letter https://doi.org/10.7554/eLife.56205.sa1
Author response https://doi.org/10.7554/eLife.56205.sa2

## Additional files

### Supplementary files

• Source data 1. Analysis of all total and polysomal mRNAs in RNAi control vs.*nsun-1*. RNAi xlsx-file containing RNA-seq analysis results for the comparison between RNAi control and *nsun-1* RNAi of total mRNAs (sheet 1) and polysomal mRNAs (sheet 2).

- Source data 2. RNA-seq analysis of transcriptome vs.translatome. xlsx-file containing RNA-seq analysis results for the comparison between total mRNAs and polysomal mRNAs for control RNAi (sheet 1) and *nsun-1* RNAi (sheet 2).

- Source data 3. Enrichment analysis of 3' UTR motifs recognized by RNA-binding proteins. xlsx-file containing the results of the enrichment analysis of 3' UTR motifs recognized by RNA-binding proteins. Differentially enriched motifs are indicated in green.

- Source data 4. Gene Ontology enrichment analysis. xlsx-file containing the results of the Gene Ontology (GO) enrichment analysis. Individual sheets represent enriched GO-terms (BP: biological process, MF: molecular function, CC: cellular component) in up- or downregulated genes in RNAi control or *nsun-1* RNAi-treated animals.

- Source code 1. R-code for RNA-seq analysis.

- Supplementary file 1. List of primers used in this study.

- Transparent reporting form

## Data availability

The raw and processed sequencing data are available from the Gene Expression Omnibus database (https://www.ncbi.nlm.nih.gov/geo) under accession GSE143618. Analyzed RNA-seq data are provided as Source data 1-4. The R-script for analyzing RNA-seq data is provided as Source code 1. Statistics for individual replicates of lifespan and stress-resistance experiments are reported in Table 1. Raw data of lifespan and thermotolerance assays are provided as Figure 2-source data 1.

The following dataset was generated:

| Author(s) | Year | Dataset title | Dataset URL | Database and Identifier |
|---|---|---|---|---|
| Schosserer M, Heissenberger C | 2020 | Translational profiling reveals that mRNAs encoding cuticle collagens are translationally repressed upon nsun-1 knockdown | https://www.ncbi.nlm. nih.gov/geo/query/acc. cgi?acc=GSE143618 | NCBI Gene Expression Omnibus, GSE143618 |

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
