## [Decision Letter]

**Acceptance summary:**

This work studies the functional consequences on cellular and organismal homeostasis of ribosomal RNA methylation. Specifically, the gene NSUN-1 can impact nematode health through its effect on rRNA methylation, providing a mechanistic link between a specific RNA modification and systemic health in a multicellular organism. To note, these ribosomal RNA modifications do not affect systemic health by altering global protein translation and have tissue/organ-specific effects, ranging from changes in body size, oocyte maturation and fecundity. Together, these findings enrich our understanding of the organismal consequences of RNA modifications.

**Decision letter after peer review:**

Thank you for submitting your article "The ribosomal RNA m^5^C methyltransferase NSUN-1 modulates healthspan and oogenesis in *Caenorhabditis elegans*" for consideration by *eLife*. Your article has been reviewed by three peer reviewers, including Dario Riccardo Valenzano as the Reviewing Editor and Reviewer #1, and the evaluation has been overseen by Jessica Tyler as the Senior Editor. The following individual involved in review of your submission has agreed to reveal their identity: Martin Sebastian Denzel (Reviewer #3).

The reviewers have discussed the reviews with one another and the Reviewing Editor has drafted this decision to help you prepare a revised submission.

As the editors have judged that your manuscript is of interest, but as described below that additional experiments are required before it is published, we would like to draw your attention to changes in our revision policy that we have made in response to COVID-19 (https://elifesciences.org/articles/57162). First, because many researchers have temporarily lost access to the labs, we will give authors as much time as they need to submit revised manuscripts. We are also offering, if you choose, to post the manuscript to bioRxiv (if it is not already there) along with this decision letter and a formal designation that the manuscript is 'in revision at *eLife*'. Please let us know if you would like to pursue this option. (If your work is more suitable for medRxiv, you will need to post the preprint yourself, as the mechanisms for us to do so are still in development.)

Heissenberge et al. study how NSUN-1 impacts rRNA methylation and health in nematodes. Eukaryotic ribosomal RNAs undergo several modifications. Among these, there are two known m^5^C, located in highly conserved target sequences. Previous work from the authors characterised the mechanism underlying one of these modifications in worms (C2381), as well as its functional consequences on cellular and organismal homeostasis. The current work focuses on the second m^5^C, at position C2982, and identifies NSUN-1 as the putative rRNA methylase. This is a novel and potentially exciting finding. Using RNAi in two worm strains, the authors show that knocking down NSUN-1 expression, the specific C2982 m^5^C level is in part (not entirely) reduced. This assay proves sufficiency (but not necessity) of NSUN-1 to reduce m^5^C levels at C2982. While it is not clear why the authors do not use a complete knock out for NSUN-1 (is it lethal?), follow-up work using RNAi explores the phenotypic effects of lowered NSUN-1 levels. While somatic and germline reduction of m^5^C levels do not have an impact on worm lifespan, it does increase resistance to heat stress, slight increase in motor activity. Reducing NSUN-1 expression separately in germline and soma showed allegedly lifespan increase. Somatic reduction of NSUN-1 leads to changes in body size, oocyte maturation and fecundity, and has no effect on global protein translation. Analysis of polysome enrichment for specific mRNAs revealed that worms with low levels of NSUN-1 have altered translation of transcripts involved in cuticle collagen deposition.

Essential revisions:

1) We are unconvinced by one of the major claims of this work, which is that C2982 has an impact on worm lifespan when expression is down in the soma. This claim does not seem to be strongly supported by the results shown. Were the replicates analysed separately or data from different assays pooled? Median lifespan appears the same between wt and RNAi worms. The survival raw data should be made available for reanalysis.

2) It is not clear whether deletion mutants for NSUN-1 (e.g. *nsun-1(tm6081)*) are viable in *C. elegans* and if yes, what is their phenotype in the context of this study. If the deletion mutant is not available, can the authors generate a CRISPR line?

3) Is there a relationship between the mRNAs selectively translated in the NSUN-1 RNAi treatment and in the NSUN-5 RNAi/mutant?

4) The results shown in Figure 1 draw a causal connection between NSUN-1 activity and C2982 based on exclusion: in other words, both NSUN-1 and NSUN-5 depletion lower the m^5^C peak by over 50%. Hence, since there are two m^5^C sites and one is written by NSUN-5, the other one must be written by NSUN-1. Is it possible that NSUN-1 may not be the only C2982 writer? Can the authors comment on this?

5) Figure 4 analyzes the gonad and oocyte maturation. While the images are very convincing, it would be good to know how penetrant the phenotype is after analysis of a larger number of animals in each group.

6) It is unclear how the observed translational remodeling that affects collagen deposition (demonstrated through the gonad extrusion and cuticle barrier phenotypes) is linked to oocyte maturation, or to heat stress resistance.

7) The authors should indicate how many times the HPLC experiments were done.

8) In Figure 3 the authors should indicate on each panel the age of the worms and at which stage the RNAi treatment was performed.

9) The overall claim about behavior should be toned down as the RNAi line has no overall improvement, but only one time point shows a difference among the groups. From the text it is not clear what statistical test was used to analyze the differences in behavior among the groups.

10) Although it may be hard to downregulate rRNAs by RNAi since they are so highly expressed, can the authors comment on whether 26S rRNA levels are reduced after RNAi and if yes to what degree?

11) While the authors write that *rrf-1* is required for amplification of the dsRNA signal specifically in the somatic tissues, this may not be completely accurate, as the Kumsta et al., 2012 paper shows that *rrf-1* affects both the soma and the germline. How does this affect the interpretation of the results?

12) Is there a chance that 26S rRNA expression or differential methylation have a tissue-specific pattern (you use RT-qPCR from whole worms)?

13) May NSUN-1 have pleiotropic effects independent of C2982 m^5^C?

---

## [Author Response]

Essential revisions:1) We are unconvinced by one of the major claims of this work, which is that C2982 has an impact on worm lifespan when expression is down in the soma. This claim does not seem to be strongly supported by the results shown. Were the replicates analysed separately or data from different assays pooled? Median lifespan appears the same between wt and RNAi worms. The survival raw data should be made available for reanalysis.

We are grateful for bringing to our attention that the presentation of pooled results, as it was done in the original manuscript, was misleading and did not convincingly support our claims. The soma-specific RNAi experiment was indeed performed in two completely independent biological replicates, which showed slight differences in the overall survival of the RNAi control. However, in each of the individual replicates an increase in mean lifespan (replicate 1: 16.5 vs. 18 days; replicate 2: 18.6 vs. 20.2 days), as well as in median lifespan (replicate 1: 16 vs. 18 days; replicate 2: 19 vs. 21 days) was clearly evident and statistically significant.

To improve the presentation of our results, we now show the first replicate as a representative experiment instead of pooled data (see revised Figure 2). This is with the exception of the thermotolerance experiments for which we performed more replicates with smaller sample sizes in order to keep the manipulation times of the animals at room temperature as short as possible. However, this rendered the individual replicates noisier and more variable. For this reason, we prefer to show the thermotolerance assay as pooled instead of a single replicate.

We now also provide a table summarizing the statistics of each individual lifespan and thermotolerance replicate as Table 1. To further increase transparency and confidence in our data, we provide, as suggested, all raw lifespan scoring data which were used to compute graphs and statistics as Figure 2—source data 1.

2) It is not clear whether deletion mutants for NSUN-1 (e.g. nsun-1(tm6081)) are viable in *C. elegans* and if yes, what is their phenotype in the context of this study. If the deletion mutant is not available, can the authors generate a CRISPR line?

We are grateful for this suggestion. We have requested the FX30263 strain from the National Bioresource Project in Japan which contains the balanced heterozygous *nsun-1(tm6081)* mutant allele. Our analysis of this strain is included in Figure 1—figure supplement 1 and the following paragraph was added in the Results section of the revised manuscript:

“In order to test if NSUN-1 is involved in large ribosomal subunit m^5^C methylation, we first sought to identify a suitable model to study loss of NSUN-1. […] One is that it allows to deplete a factor of interest at a particular life stage only (e.g. in adult worms), another is that it allows performing tissue-specific knockdown of gene expression.”

We agree that, in principle, generating additional mutant strains is always interesting. Regretfully, the timeframe involved in generating such CRISPR-Cas9 mutants in worm, and, in particular a knock-in, is simply not compatible with the revision of the present work, and we feel it should thus be left for future work.

3) Is there a relationship between the mRNAs selectively translated in the NSUN-1 RNAi treatment and in the NSUN-5 RNAi/mutant?

To address this point, we included an additional figure (Figure 6), as well as the following description in the Results section:

“To test this possibility, we isolated mRNAs contained in the polysomal fraction, systematically sequenced them by RNA-seq and compared their abundance in polysomes between animals subjected to *nsun-1* RNAi versus RNAi control. […] This suggests that the pattern of translated mRNAs upon *nsun-1* depletion is highly specific and strikingly distinct from that observed upon loss of the other 26S rRNA m^5^C methyltransferase.”

We would like to add that a recent publication reported that loss of N6-adenosine methylation of 18S rRNA by METL-5 depletion also caused a reprogramming of the translatome in *C. elegans*. However, the observed changes in translated mRNAs were different from those reported by us for NSUN-1 (Liberman et al., https://doi.org/10.1126/sciadv.aaz4370). This further emphasizes the relevance and timeliness of our study and implicates that modulation of different rRNA methylations clearly promote distinct translational responses.

4) The results shown in Figure 1 draw a causal connection between NSUN-1 activity and C2982 based on exclusion: in other words, both NSUN-1 and NSUN-5 depletion lower the m^5^C peak by over 50%. Hence, since there are two m^5^C sites and one is written by NSUN-5, the other one must be written by NSUN-1. Is it possible that NSUN-1 may not be the only C2982 writer? Can the authors comment on this?

We included the following paragraph in the Results section of the revised manuscript:

“Since our conclusion is based on depletion of *nsun-1* to ~20% residual expression and not on a full gene knockout (Figure 1—figure supplement 2), we cannot exclude the formal possibility that NSUN-1 might not be the only m^5^C2982 writer in *C. elegans*. However, we consider this possibility to be highly unlikely because the combination of a knockout of Rcm1 with a catalytic mutation of Nop2 in yeast was sufficient to completely remove m^5^C from 25S rRNA (Sharma et al., 2013).”

5) Figure 4 analyzes the gonad and oocyte maturation. While the images are very convincing, it would be good to know how penetrant the phenotype is after analysis of a larger number of animals in each group.

Following the reviewer’s suggestion, we analyzed >50 worms per group (RNAi control vs. *nsun-1* RNAi) of the SA115 and JJ1473 strains and found that the phenotype is 100% penetrant. We did not find a single worm showing normal gonad morphology after exposure to *nsun-1* RNAi. We included this information in the Results section, the legend of Figure 4 and the legend of Figure 4—figure supplement 1.

6) It is unclear how the observed translational remodeling that affects collagen deposition (demonstrated through the gonad extrusion and cuticle barrier phenotypes) is linked to oocyte maturation, or to heat stress resistance.

We agree that this remains elusive at this point and will not be elucidated here.

We did hypothesize that the altered abundance of transcripts containing 3’ UTR motifs, which are recognized by important translational regulators of development such as GLD-1, are key to the defects in oocyte maturation upon *nsun-1* depletion. We therefore exposed a GDL-1::GFP reporter strain to *nsun-1* RNAi and made the following observations, which we now included in the Results section:

“Indeed, GLD-1::GFP protein expression was restricted exclusively to a small portion of the loop region in adult nematodes (Figure 7—figure supplement 1). Taken together, these findings suggest that *nsun-1* is required for correct gonadal GLD-1 localization during development, which might then directly or indirectly influence the specific translation of mRNAs required for further steps in development.”

However, no obvious and direct link between GLD-1 function, collagen deposition and heat stress resistance is evident from our RNA-seq data, since differentially translated collagen genes did not show an enrichment of GLD-1 binding motifs. Since evidence suggests that loss of collagen genes like *col-35* contributes to brood size regulation (Rual JF et al., 2004; and Ceron J et al., 2007), we might speculate that the altered translation of collagens is sensed by a factor upstream of GDL-1, which then inhibits oozyte maturation further downstream. Due to the vast number of differentially regulated collagens, we feel that systematically testing this hypothesis in epistasis experiments is out of scope of the present study.

A recent publication by Liberman et al. (https://doi.org/10.1126/sciadv.aaz4370), together with our data, implies that improved thermotolerance might be a common phenotype upon inhibition of specific rRNA modifications. The underlying mechanisms, however, appear to be distinct. cyp-29A3, which modulates heat stress resistance upon *metl-5* knockdown (Liberman et al., https://doi.org/10.1126/sciadv.aaz4370) , was not differentially expressed in our dataset upon *nsun-1* knockdown (log2 fold-change in the translatome of *nsun-1* RNAi vs. control RNAi = 0.24, adjusted p-value = 1). We can therefore only speculate that protein maintenance by chaperons or protein degradation by the catabolic arm of proteostasis (autophagy and ubiquitin/proteasome system) might be sensitive to slight alterations in translation, as imposed by loss of a single rRNA methylation. These pathways might then be rendered more responsive to acute stress and thereby mediate enhanced survival by hormesis. This or alternate hypotheses remain to be systematically tested in future studies.

7) The authors should indicate how many times the HPLC experiments were done.

We independently performed all HPLC experiments twice. Thus, we exposed three different strains two times to *nsun-1* RNAi and found a clear reduction in m^5^C levels of 26S rRNA in all six cases. Both independent experiments are presented in the new Figure 1—figure supplement 3. We also included this information in the figure legend of Figure 1.

8) In Figure 3 the authors should indicate on each panel the age of the worms and at which stage the RNAi treatment was performed.

We thank the reviewer for this suggestion and included the relevant information in Figure 3, as well as in the respective figure legend.

9) The overall claim about behavior should be toned down as the RNAi line has no overall improvement, but only one time point shows a difference among the groups. From the text it is not clear what statistical test was used to analyze the differences in behavior among the groups.

We agree with the reviewer and have toned down the text as requested, removing any strong statements about an improvement of healthspan. Statistical significance at each timepoint was determined using multiple comparison adjusted t-tests by the Holm-Sidak method. We included this information in the legend of Figure 2 and indicated statistical significance in the figure with asterisks. As noticed by the reviewer, *nsun-1* depletion significantly increased locomotion only at day 8, while nsun-5 RNAi increased the average speed at day 12.

10) Although it may be hard to downregulate rRNAs by RNAi since they are so highly expressed, can the authors comment on whether 26S rRNA levels are reduced after RNAi and if yes to what degree?

Considering Figure 5C where we loaded equal amounts of total RNA on a gel and quantified the 26S and 18S rRNA band, we did not observe a reduction of 26S rRNA relative to 18S rRNA or total RNA upon *nsun-1* RNAi in any of the three different genetic backgrounds. However, knockout of nsun-5 (JGG1 strain) reduced 26S and 18S rRNA levels by 50% and 46%, respectively.

11) While the authors write that rrf-1 is required for amplification of the dsRNA signal specifically in the somatic tissues, this may not be completely accurate, as the Kumsta et al., 2012 paper shows that rrf-1 affects both the soma and the germline. How does this affect the interpretation of the results?

The reviewer correctly remarks that Kumsta and coworkers reported in 2012 that *rrf-1* mutations maintain RNAi efficiency in the soma in addition to the germline. However, this does not affect interpretation of our results, since we neither found a change in lifespan or animal size in the *rrf-1* mutant strain NL2098, nor in N2 wild-type animals. The reported changes in lifespan and size were only present in soma-specific NL2550 animals.

Zou and coworkers introduced recently the novel DCL569 strain (Zou et al.,2019), which shows improved germline-specificity compared to NL2098. To rule out any effect of residual somatic RNAi activity in NL2098, we repeated the length measurement upon *nsun-1* RNAi and nsun-5 RNAi in the DCL569 strain and did not find a difference compared to control RNAi, confirming our previous experiment (Figure 3—figure supplement 1).

12) Is there a chance that 26S rRNA expression or differential methylation have a tissue-specific pattern (you use RT-qPCR from whole worms)?

The reviewer raises a very interesting question here, which will be fascinating to elucidate in the future. At tissue level in mice and humans, only few RNA modifications are known to show plasticity in response to environmental changes. We would expect that more pronounced alterations might happen at single-cell level, which we are currently unable to investigate due to limited sensitivity of available methods.

*C. elegans* might provide exciting opportunities in this regard, since sufficient amounts of RNA might be easily generated from a genetically homogenous population of animals. If methods to purify ribosomes from individual tissues of nematodes become available, it would be technically feasible and exciting to determine tissue-specific rRNA modification patterns, and their corresponding translatomes.

13) May NSUN-1 have pleiotropic effects independent of C2982 m^5^C?

We consider it very likely the NSUN-1 carries additional effects independent of its methylation activity. We included the following paragraph in the Discussion:

“Accumulating data suggests that NSUN-1 has pleiotropic effects independent of its methylation activity. [...] Similarly, deletion of Nop2, the yeast homolog of *nsun-1*, was shown to be lethal, but viability could be restored by re-expression of a catalytic mutant (Sharma et al., 2013).”